# FGF4 initiates X-chromosome inactivation via activating YY1 and prompting loss of pluripotency factors

Lizhu Ma[1,3], Wei Fu[1,2,3], Lei An[1,3], Qianying Yang[1], Ruiqi Hai [ID][1], Guangyin Xi[1], Yinjuan Wang[1], Juan Liu[1], Chao Zhang[1], Yao Fu[1], Zhenni Zhang[1], Xiaodong Wang[1] & Jianhui Tian [ID][1✉]

## Abstract

X-chromosome inactivation (XCI) balances gene expression between sexes in mammals and is essential to female development. XCI initiation strictly relies on the upregulation of long noncoding RNA *Xist* upon differentiation. Despite the co-occurrence and tight correlation between XCI and differentiation, master coordinators to synchronize XCI and differentiation remain ill-defined. Here, we report that FGF4, an autocrine differentiation-prompting stimulus, is essential for *Xist* upregulation and XCI initiation in mouse embryonic stem cells (ESCs). Either *Fgf4* deficiency or FGFR blocking results in failure of *Xist* upregulation and XCI initiation. Mechanistically, FGF4 initiates XCI in a MEK/ERK-dependent manner, via two parallel but opposing pathways: i)FGF4 phosphorylates and activates YY1, a robust transcription activator of *Xist*, and ii) FGF4 facilitates decline of pluripotency factors *Prdm14*, *Nanog* and *Rex1*, resolving *Xist* repression. Together, we show how FGF4 comprehensively orchestrates XCI and ESC differentiation, and ensures XCI initiation by coordinating two opposing regulators that directly influence *Xist* transcription. The FGF-ERK-YY1 axis also constitutes a missing link between ubiquitously expressed *Yy1* and its functional activation responsible for *Xist* upregulation and XCI initiation.

**Keywords** X-chromosome Inactivation; FGF4; MEK/ERK; YY1; Embryonic Stem Cells
**Subject Categories** Chromatin, Transcription & Genomics; Development; Stem Cells & Regenerative Medicine

## Introduction

X chromosome inactivation (XCI) is a crucial developmental process that has evolved in eutherian mammals to enable X-chromosome dosage balance between males (XY) and females (XX) by transcriptionally silencing one of the two X chromosomes in females (Galupa and Heard, 2015; Wutz, 2011). In mice, two forms of XCI occur during early development. Imprinted XCI, which preferentially silences the paternal X chromosome (Xp), initiates in early preimplantation embryos and is maintained in extraembryonic tissues but lost in the inner cell mass (ICM). Shortly after this, the random form of XCI, which silences either the maternal X chromosome (Xm) or the Xp, takes place in the ICM-derived differentiating epiblast or embryonic stem (ES) cells (Augui et al, 2011; Lee, 2011). Initiation of random XCI strictly relies on the upregulation of long noncoding RNA *Xist* upon differentiation from the pluripotent state(Penny et al, 1996), during which the loss of pluripotency derepresses *Xist*(Augui et al, 2011). However, whether and how differentiation-promoting factors can positively regulate XCI initiation remains unclear.

Given the spatiotemporal co-occurrence and tight correlation between XCI and differentiation, an unknown master coordinator must therefore be responsible for synchronizing these two processes, at the top of a complex regulatory network controlling *Xist* expression.

The potential master coordinator should have some properties including: (i) differentiation-promoting effect because random XCI is only initiated upon the differentiation of ES cells or epiblast; (ii) stimulating effect on activating factors controlling *Xist* expression because robust *Xist* transcriptional activation is the prerequisite for the onset of XCI (Brockdorff et al, 1991; Penny et al, 1996); (iii) likely to be an autocrine factor secreted by the ICM-derived differentiating epiblast or ES cells given XCI initiation is a cell-autonomous process (Del et al, 2017; Pacini et al, 2021).

FGF4 caught our attention because it is a well-known autoinductive differentiation-prompting signal for ES cells to exit the pluripotent state and peri-implantation embryogenesis progresses (Feldman et al, 1995; Kunath et al, 2007; Molotkov et al, 2017). A short pulse of exogenous FGF4 rescued *Fgf4*⁻/⁻ ES cell differentiation (Molotkov et al, 2017), suggesting that endogenous FGF4 produced by ES cell is needed as a permissive autocrine signal to allow the initiation of ES cell differentiation. Moreover, we

[1]Frontiers Science Center for Molecular Design Breeding (MOE), State Key Laboratory of Animal Biotech Breeding, Key Laboratory of Animal Genetics, Breeding and Reproduction of the Ministry of Agriculture and Rural Aifairs, National Engineering Laboratory for Animal Breeding, College of Animal Science and Technology, China Agricultural University, Beijing, People's Republic of China. [2]Present address: Key Laboratory of Qinghai-Tibetan Plateau Animal Genetic Resource Reservation and Utilization, Southwest Minzu University, Ministry of Education, Chengdu, China. [3]These authors contributed equally: Lizhu Ma, Wei Fu, Lei An. ✉E-mail: tianjh@cau.edu.cn

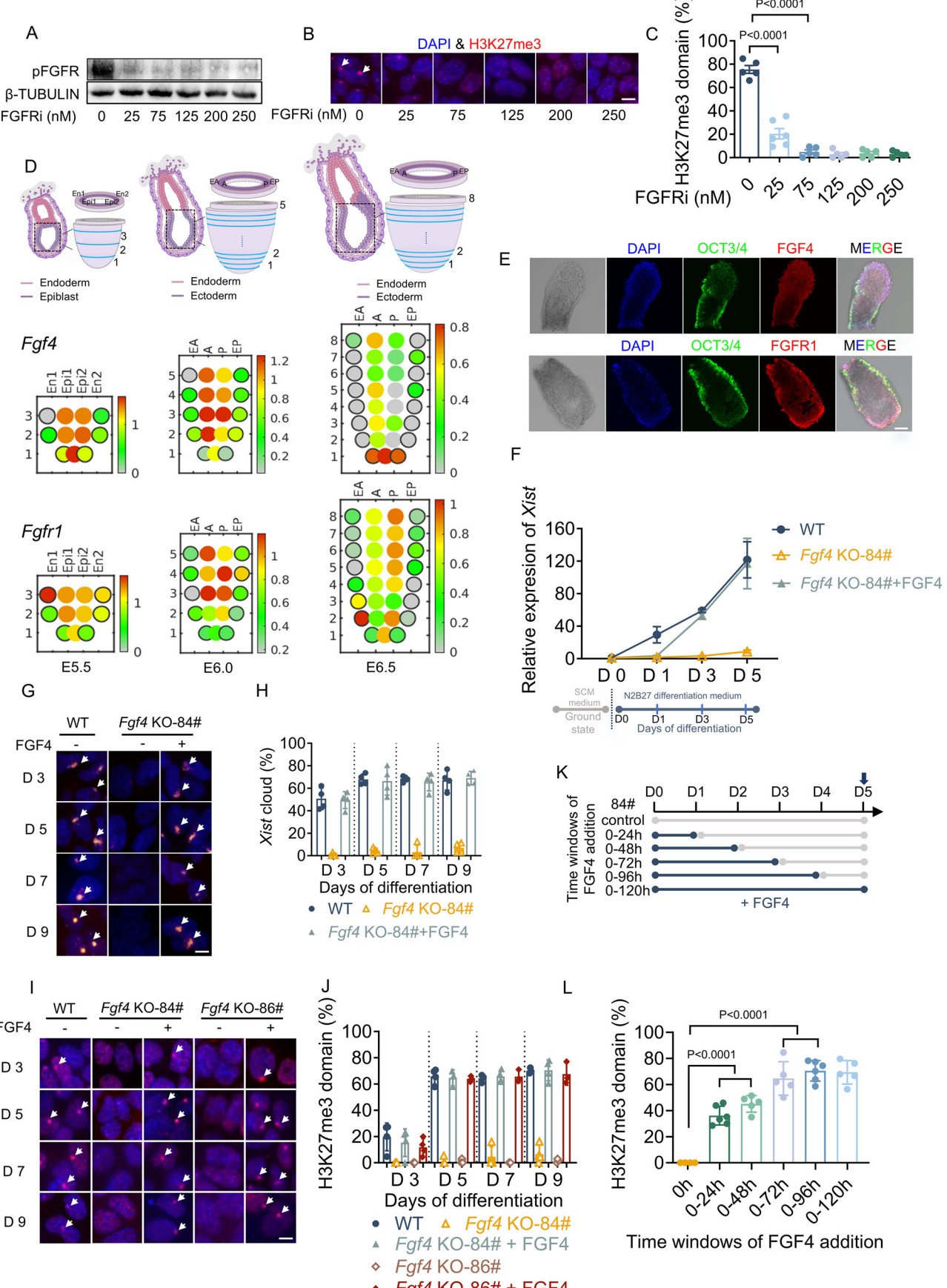

**Figure 1. FGF4 is essential for upregulating *Xist* expression and initiating random XCI.**

(A) The protein level of pFGFR in WT ES cells after addition of 0, 25, 75, 125, 200, and 250 nM pan-FGFR inhibitor (BGJ398) on day 5 of differentiation. (B, C) Immunofluorescence staining (B) and quantification (C) of H3K27me3 domains in WT, *Fgf4* KO-84#, *Fgf4* KO-84# + FGF4 ES cells on day 5 of differentiation during BGJ398 treatment. $n = 5$–6; scale bars: 10 µm; FGF4: 10 ng/mL. The white arrow denotes a nuclear H3K27me3 domain. % H3K27me3 = number of cells with nuclear H3K27me3 domains/total number of analyzed cells. Cells with one H3K27me3 domain were considered as normal XCI and included in the statistical unit. (D) The presentation of the spatial pattern of gene expression of *Fgf4* and *Fgfr1* in gastrulating mouse embryos. The upper panel: the schematic diagram of E5.5, E6.0, and E6.5 embryos. The boxed region indicates the epiblast, and the schematic model (right) depicts the sample section and captured regions. The position of each sectional sample is labeled by the number on the distal-proximal axis. The captured regions are labeled in the section, and the corresponding abbreviations in each section are indicated below. The lower panel: spatial expression pattern of *Fgf4* and *Fgfr1* in the epiblast at the indicated stage, which is presented as the corn plot (so named for its resemblance to a corn cob). Each dot in the plot represents the cell sample at the specific positional address indicated by the number next to the plot and the abbreviation at the top. The color indicates the level of gene expression computed from the transcript counts in the 3D gene expression database of gastrulating mouse embryos. Data source: eGastrulation (http://egastrulation.sibcb.ac.cn). (E) Immunofluorescence staining of FGF4 and FGFR1 in E6.5 epiblasts; scale bars: 75 µm. OCT3/4 was used as the marker specific to the epiblast. (F) Dynamic expression levels of *Xist* in WT, *Fgf4* KO-84#, and *Fgf4* KO-84# + FGF4 ES cells on day 0, 1, 3, and 5 of differentiation. Data were represented as means ± SEM; $n \geq 6$. The schematic diagram below the X-axis indicates the timeline of differentiation. The day when ES cells were transferred to the differentiation medium was designated as day 0. (G, H) FISH staining (G) and quantification (H) of *Xist* clouds in WT, *Fgf4* KO-84#, *Fgf4* KO-84# + FGF4 ES cells on day 3, 5, 7, and 9 of differentiation; $n = 4$; scale bars: 10 µm. The white arrow denotes a nuclear *Xist* cloud. % *Xist*-cloud = number of cells with *Xist* cloud/total number of analyzed cells. Cells with one *Xist* cloud signal were considered normal XCI and included in the statistical unit. (I, J) Immunofluorescence staining (I) and quantification (J) of H3K27me3 domain in WT, *Fgf4* KO-84#, *Fgf4* KO-84# + FGF4 ES cells on day 3, 5, 7, and 9 of differentiation; $n = 3$–4; scale bars: 10 µm. The white arrow denotes a nuclear H3K27me3 domain. % H3K27me3 = number of cells with nuclear H3K27me3 domains/total number of analyzed cells. Cells with one H3K27me3 domain were considered as normal XCI and included in the statistical unit. (K, L) Immunofluorescence staining (K) and quantification (L) of H3K27me3 domains in *Fgf4* KO-84# after FGF4 addition at different time windows; $n = 4$–6. Control, cells without FGF4 supplement during the process of differentiation; 0–24, 0–48, 0–72, 0–96, and 0–120 h, cells are treated with FGF4 for 24, 48, 72, 96, and 120 h, respectively, upon differentiation, then transferred to the differentiation medium without FGF4 addition until the H3K27me3 staining on day 5 of differentiation. (C, L) Data are shown as means ± SEM. *P* value was calculated by one-way ANOVA test with multiple comparisons. For more details, please see the "Methods". Source data are available online for this figure.

postulated that the FGF4–ERK–YY1 axis may exist and act to upregulate *Xist*. YY1 is an essential transcription factor that directly triggers robust *Xist* upregulation at the onset of XCI (Makhlouf et al, 2014). However, some uncertainties remain unexplained. As a ubiquitously expressed general transcription factor, YY1 maintains the steady-state levels upon XCI initiation and binds *Xist* regulatory regions in undifferentiated ES cells, which express only low levels of *Xist* (Gontan et al, 2012; Makhlouf et al, 2014). These facts are reminiscent of the missing link between XCI initiation and functional activation of YY1. Previous studies have indicated that FGF signaling or MEK-ERK could facilitate expression (Ji et al, 2015) and nuclear translocation of YY1 (Stoeckius et al, 2012). In addition, ERK-mediated phosphorylation of YY1 was thought to regulate its DNA-binding activity, thus modulating the transcriptional activation of target genes (He et al, 2010; Martinez-Moreno et al, 2017). However, despite evidence implicating the role of FGF in increasing *Xist* expression in female ground-state iPS cells and primed ES cells (An et al, 2020; Di Stefano et al, 2010; Sarel-Gallily and Benvenisty, 2022; Wang et al, 2011), whether and how FGF4 can act as a key upstream regulator that initiates *Xist* upregulation remains to be determined.

In our study, using loss- and gain-of-function experiments under the standard XCI study model, we dissected that autocrine FGF4 is the top signaling of *Xist* upregulation and XCI initiation. In *Fgf4⁻/⁻* ES cells, XCI initiation almost totally failed. The exogenous addition of FGF4 completely rescued this failure. We further showed FGF4 drives XCI in a MEK/ERK-dependent manner. Two parallel but opposing pathways that can respond to FGF signaling, i.e., phosphorylation of YY1 and decline of pluripotency factors, synergistically act to upregulate *Xist*. Thus, our results not only advance the current knowledge about the role of *FGF4*, as a key coordinator, in balancing the regulatory network of hallmark epigenetic events, but also answer how each embryonic cell autonomously controls XCI.

# Results

## FGF4 is essential for upregulating *Xist* expression and initiating random XCI

To examine the crucial role of FGF signaling in the initiation and occurrence of random XCI, we initially employed female mouse ES cells, a well-established ex vivo autonomous model for investigating random XCI (Tian et al, 2010) that helps eliminate any influence from maternal factors (Del et al, 2017). We exposed female ES cells to Infigratinib (BGJ398), a pan-FGFR inhibitor, and detected H3K27me3 state, the hallmark of XCI, at day 5 of differentiation. Immunofluorescent staining results revealed that blockage of FGF signaling led to a nearly complete failure of random XCI (Fig. 1A–C).

Then, we detected if FGF4 is the prominent FGF ligand during the developmental window of random XCI initiation, using the online tool (http://egastrulation.sibcb.ac.cn) that profiles the 3D transcriptome of the mouse embryo from pre-gastrulation (embryonic day (E5.5) to late gastrulation (E6.5). Using the online tool (eGastrulation: http://egastrulation.sibcb.ac.cn) of the 3D gene expression database of gastrulating mouse embryos, we reconstructed spatiotemporal expression patterns of FGF family members and their interacting receptors. We noticed that *Fgf4*, *Fgf5*, and *Fgf8* (Fig. 1D; Appendix Fig. S1A). were highly expressed in the embryonic epiblast. Our focus, *Fgf4* and its interacting receptor *Fgfr1* (Fig. 1E), showed high expression levels in the epiblast. Moreover, reanalysis results of different transcriptome data, along with the real-time quantitative PCR (RT-qPCR) analysis, consistently showed that *Fgf4* is the primary expressed ligand compared to other ligands. Both *Fgf4* and its major receptor *Fgfr1* are maintained at steady-state levels in the early phases of ES differentiation, by which rXCI initiates. Calculation of FGF4-FGFR1 interaction scores also indicated that FGF4 signaling is highly active during the stage (Appendix Fig. S1B–D).

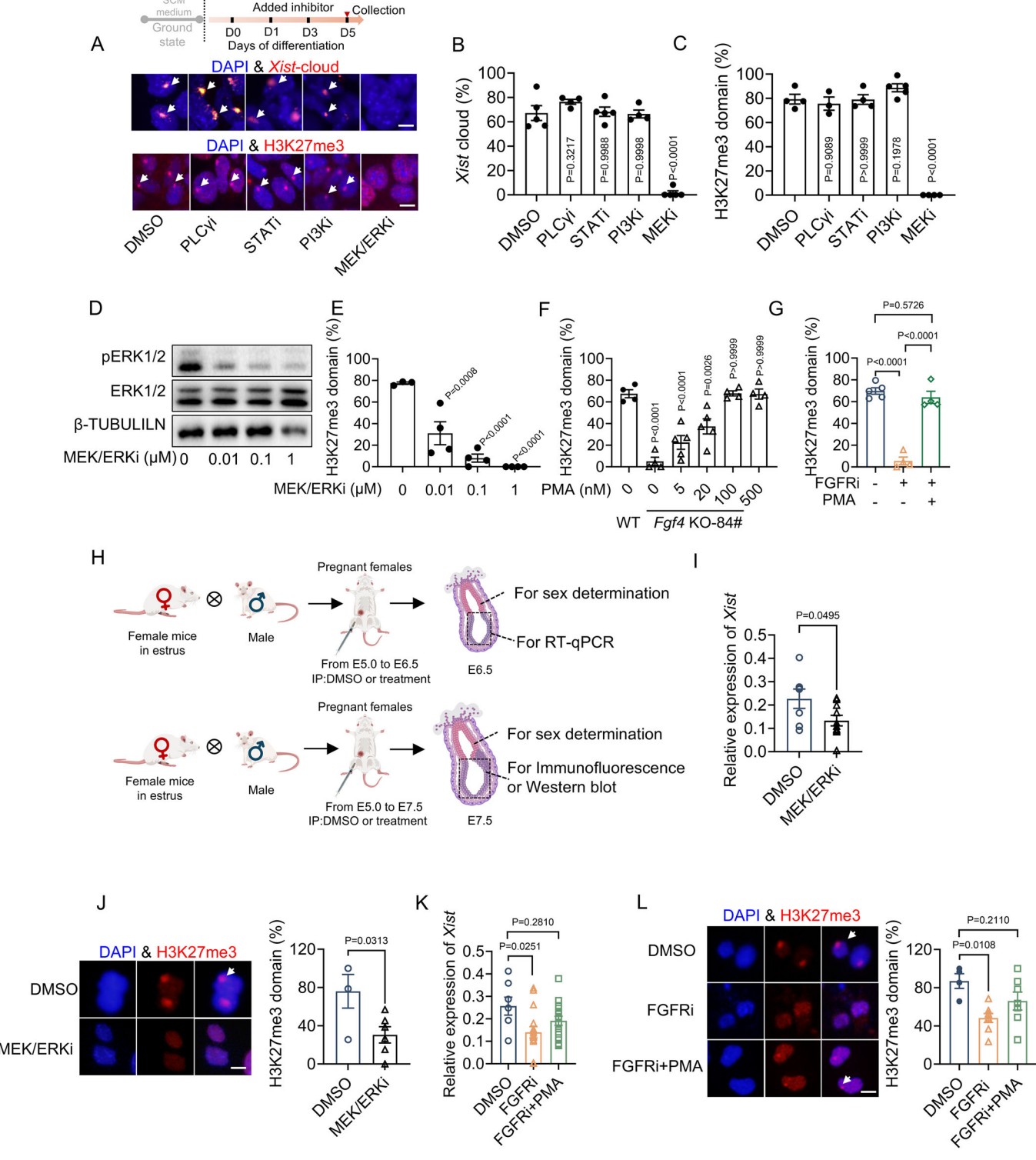

We next attempted to know how autocrine FGF4 facilitates random XCI initiation. CRISPR-Cas9n–mediated functional knockout of *Fgf4* (Appendix Fig. S2A–D) resulted in a complete failure of *Xist* upregulation and a strong decrease in the number of cells displaying *Xist* RNA cloud, as revealed by RT-qPCR and RNA FISH analyses (Fig. 1F–H), both of which are the prerequisites for initiating

XCI (Makhlouf et al, 2014). In line with this, *Fgf4*−/− ES cells showed a nearly complete failure of random XCI: time-course immunofluorescent staining analysis of the H3K27me3 state revealed that random XCI was kept at nearly undetectable levels in differentiated *Fgf4*−/− ES cells until day 9 of differentiation. This finding was reproduced with two independent knockout cell lines (*Fgf4* KO-84#, 86#) (Fig. 1I,J). Of

**Figure 2. FGF4 initiates random XCI in a MEK/ERK-dependent manner.**

(A) Schematic of inhibitor treatment during differentiation from D0 to D5 (upper panel), and Immunofluorescence staining of H3K27me3 and *Xist* FISH staining (down panel) in WT ES cells treated with the inhibitor specific to PLCγ, STAT, PI3K, and MEK/ERK on day 5 of differentiation. (B, C) Immunofluorescence staining of H3K27me3 and *Xist* FISH staining, and corresponding quantification of H3K27me3 domains (B) and *Xist* clouds (C) in WT ES cells treated with the inhibitor specific to PLCγ, STAT, PI3K, and MEK/ERK on day 5 of differentiation; $n = 3–5$ for B; $n = 4–5$ for (C); scale bars: 10 µm. PLCγi: U73122 (U), 5 µM; STATi: Fludarabine (Flu), 1 µM plus SH-4-54 (SH), 1 µM; PI3Ki: LY294002 (LY) 1 µM; MEK/ERKi: PD0325901 (PD), 1 µM. In the upper panels, the white arrow in the upper panels denotes a nuclear *Xist* cloud. % *Xist*-cloud = number of cells with *Xist* cloud/total number of analyzed cells. Cells with one *Xist* cloud signal were considered as normal XCI and included in the statistical unit. In the lower panels, the white in the nuclear H3K27me3 domain. % H3K27me3 = number of cells with nuclear H3K27me3 domains/ total number of analyzed cells. Cells with one H3K27me3 domain were considered as normal XCI and included in the statistical unit. (D) Western blot analysis of ERK and pERK in WT ES cells treated with 0.01, 0.1, and 1 µM MEK/ERK inhibitor (MEK/ERKi) on day 5 of differentiation. (E) Quantification of H3K27me3 domains in WT ES cells treated with 0.01, 0.1, and 1 µM MEK inhibitor on day 5 of differentiation; $n = 3–4$. A two-tailed unpaired Student's *t* test was used to assess the differences between the control and experimental groups. (F) Quantification of H3K27me3 domains in WT and *Fgf4* KO-84# ES cells treated with or without 5, 20, 100, and 500 nM PMA on day 5 of differentiation; $n = 4–5$. (G) Quantification of H3K27me3 domains in WT ES cells subject to FGFR inhibition or in combination with PMA-induced ERK activation on day 5 of differentiation; $n = 4–5$. FGFRi: BGJ398, 75 nM; PMA: 100 nM. (H) Schematic diagram of the experimental workflow. Pregnant mice received daily intraperitoneal injections of the MEK/ERK inhibitor, FGFR inhibitor, or in combination with the ERK-activating agent PMA, during the window of random XCI. The female epiblast was collected and tested for *Xist* expression and iXCI status. (I) Relative expression levels of *Xist* in in vivo E6.5 epiblast from females undergoing intraperitoneal injections with DMSO, MEK/ERKi; $n = 7–10$. MEK/ERKi: PD0325901, 10 mg/kg. (J) Immunofluorescence staining (left) and quantification (right) of H3K27me3 domains in in vivo E6.5 epiblasts from females undergoing intraperitoneal injection with DMSO, MEK/ERKi; $n = 4–6$; scale bars: 10 µm. The white arrow denotes a nuclear H3K27me3 domain. % H3K27me3 = number of cells with nuclear H3K27me3 domains/total number of analyzed cells. Cells with one H3K27me3 domain were considered as normal XCI and included in the statistical unit. (K) Relative expression levels of *Xist* in in vivo E6.5 epiblast from females undergoing intraperitoneal injection with DMSO, FGFRi, and FGFRi+PMA, respectively; $n = 7–17$. FGFRi: BGJ398, 10 mg/kg; PMA: 2.5 mg/kg. (L) Immunofluorescence staining (left) and quantification (right) of H3K27me3 domains in in vivo E6.5 epiblasts from females undergoing intraperitoneal injection of DMSO, MEK/ERKi, FGFRi and FGFRi+PMA, respectively; $n = 4–8$; scale bars: 10 µm. The white arrow denotes a nuclear H3K27me3 domain. % H3K27me3 = number of cells with nuclear H3K27me3 domains/total number of analyzed cells. Cells with one H3K27me3 domain were considered as normal XCI and included in the statistical unit. (B, C, E–G, I–L). Data are shown as means ± SEM. *P* value was calculated by one-way ANOVA test with multiple comparisons (B, C, E–G, K, L) and unpaired two-tailed Student's *t* test (I, J). For more details, please see the "Methods". Source data are available online for this figure.

note, loss of H3K27me3 and *Xist* domains is not due to the loss of the X chromosome per se or the health status of *Fgf4* KO-84# ES cells, as revealed by karyotype analysis and flow cytometric analysis of apoptosis (Appendix Fig. S2D–G). More importantly, we found that the addition of exogenous FGF4 to the differentiation medium, which compensates for the endogenous loss of FGF4, completely rescued *Xist* upregulation and coating, as well as the H3K27me3 state, to levels comparable to those in wild-type ES cells (Fig. 1I,J). The beneficial effect of FGF4 in promoting random XCI showed a dose-dependent manner and had a synergistic effect with heparin, a cofactor that is required for activating FGF4 and its binding to receptors (Appendix Fig. S3A,B). We next used an alternative, but non-physiological FGF ligand, FGF2, as it is not expressed in the ICM (Kang et al, 2017) and ground-state ES cells (Appendix Fig. S1B,C). A previous study used FGF2 to restore *Fgf4* loss-induced deficiency in primitive endoderm (PrE) lineage differentiation, or mimic the function of FGF4 in driving PrE differentiation (Kang et al, 2013; Molotkov et al, 2017). When using FGF2, we obtained a similar result of completely rescuing the random XCI (Appendix Fig. S3C). Moreover, we further indicated that even in an undifferentiated ground state, FGF4 significantly facilitated *Xist* expression (Appendix Fig. S3D). These data strongly suggest that *Xist* upregulation and random XCI initiation are highly sensitive to FGF signaling.

Given *Fgf4* was highly expressed during the initiation window of random XCI, we postulated that FGF4-induced random XCI initiation is restricted to early stages of differentiation. Using FGF4 supplementation experiments with different time intervals, we found that the addition of FGF4 from day 0 to day 3 is enough to irreversibly trigger random XCI initiation in differentiated *Fgf4* KO-84# ES cells (Fig. 1K,L). By contrast, adding FGF4 from day 3 onwards, even for 3 days, failed to fully initiate random XCI (Appendix Fig. S3E), suggesting that FGF4 has a clear functional window in initiating random XCI.

## FGF4 initiates random XCI in a MEK/ERK-dependent manner

To decipher the mechanism responsible for FGF4-induced *Xist* expression and random XCI, we then examined the canonical downstream pathways that mediate FGF signaling. Only inhibiting the MEK/ERK pathway, but not other pathways, could recapitulate the complete loss of *Xist* clouds and H3K27me3 domains observed in *Fgf4* KO-84# ES cells, as revealed by RNA FISH and immunofluorescent staining (Fig. 2A–C; Appendix Fig. S4A). Furthermore, the degree of ERK inhibition was inversely correlated with the occurrence of random XCI, indicating that random XCI is greatly influenced by ERK activity (Fig. 2D,E). In line with this, either FGFR inhibition or *Fgf4* knockout resulted in reduced ERK activation during the initiation phase of ES cell differentiation (Appendix Fig. S4B). When we added exogenous FGF4, the ERK pathway was reactivated (Appendix Fig. S4C), which in turn rescued *Xist* upregulation and H3K27me3 state (Fig. 1F,I,J). More importantly, in *Fgf4* KO-84# ES cells, either PMA-induced ERK reactivation or direct overexpression of constitutively active Mek (*CaMek1*) could rescue the random XCI state, to a level comparable to those in wild-type ES cells (Fig. 2F,G; Appendix Fig. S4D,E), indicating that ERK activation is essential and sufficient for initiating random XCI in ES cells.

These observations were further confirmed by the results from the in vivo experiments. Because both male and female *Fgf4*−/− embryos exhibited severe defects or even complete lethality during implantation and early gastrulation (Feldman et al, 1995), a stage-specific chemical strategy was used. Pregnant mice received daily intraperitoneal injections of the MEK/ERK inhibitor, FGFR inhibitor, or in combination with ERK-activating agent PMA, during the window of random XCI. The female epiblast was collected and tested for *Xist* expression and iXCI status (Fig. 2H).

ERK inhibition significantly impeded *Xist* upregulation and reduced H3K27me3 domains in the epiblast (Fig. 2I,J; Appendix Fig. S4F). Importantly, FGFR inhibition-induced XCI failure could be directly rescued by PMA-induced ERK activation (Fig. 2K,L; Appendix Fig. S4G). Collectively, both in vivo and ex vivo loss- and gain-of-function experiments indicated that FGF4 initiates random XCI in a MEK/ERK-dependent manner.

## FGF4 regulates Xist upregulation via the ERK–YY1 axis

Having confirmed the role of MEK/ERK in mediating FGF4-induced *Xist* expression and random XCI, we next investigated mechanisms downstream of ERK activation. Because we reasoned that FGF4 should stimulate positive regulators of *Xist*, we focused on the known XCI regulators that could be the potential targets of ERK. YY1, an essential transcription factor that directly triggers robust *Xist* upregulation at the onset of XCI (Makhlouf et al, 2014), is of great interest to us. Previous studies have indicated that FGF signaling or MEK/ERK could facilitate expression (Ji et al, 2015), nuclear translocation, and DNA-binding activity of YY1 (He et al, 2010; Martinez-Moreno et al, 2017; Stoeckius et al, 2012). These properties may fill the missing link between ubiquitously expressed *Yy1* and its functional activation responsible for Xist upregulation and XCI initiation, i.e., YY1 maintains the steady-state levels upon XCI initiation and binds *Xist* regulatory regions in undifferentiated ES cells, which express only low levels of *Xist* (Makhlouf et al, 2014). Analyses of RT-qPCR, as well as Western blot and immunofluorescent staining, showed that neither YY1 expression levels nor its nuclear localization was affected by FGF4 deficiency and exogenous addition (Appendix Fig. S5A–C).

In contrast, Phos-tag-based Western blot showed that YY1 phosphorylation levels, above the basal level, were transiently increased at the window of differentiation initiation, approximately in line with the ERK phosphorylation activation during the initial differentiation (Fig. 3A). In contrast, under same differentiation condition, male ES cells (R1) displayed a much weaker change in YY1 phosphorylation than that in female PGK12.1 ES cells, suggesting that the increased YY1 phosphorylation during early stage of differentiation may be prone to occur in female ES cells (Appendix Fig. S5D,E). Moreover, the phosphorylated levels of both YY1 and ERK, as well as their interaction revealed by coimmunoprecipitation (co-IP) assay at day 1 of differentiation, were significantly reduced in *Fgf4* KO-84# ES cells, while exogenous FGF4 addition considerably rescued YY1 phosphorylation (Fig. 3B–D). Similarly, inhibiting ERK in wild-type ES cells recapitulated the decrease in YY1 phosphorylation (Fig. 3E), indicating that YY1 phosphorylation depends on FGF4-induced ERK activation.

We next asked whether ERK-dependent phosphorylation of YY1 affects its binding to the *Xist* promoter. FGF4 knockout or ERK inhibition-induced decrease in YY1 phosphorylation reduced the enrichment of YY1 to specific motifs located at the *Xist* promoter (Fig. 3F). In contrast, exogenous FGF4 addition significantly rescued the enrichment of YY1 in *Fgf4* KO-84# ES cells (Fig. 3G). In addition to *Xist* promoter, CUT&Tag analysis revealed that FGF4 deficiency leads to extensive genome-wide loss of YY1 binding (Fig. 3H–J; Appendix Fig. S5F,G), which has been also confirmed by CUT&RUN-qPCR (Appendix Fig. S5H).

To further confirm the functional role of YY1 phosphorylation on *Xist* transcription at the onset of random XCI, we next mutated either serine (Ser) 120 or 247, two identified potential phosphorylated residues of YY1 (Fig. 3K) (He et al, 2010; Martinez-Moreno et al, 2017) to alanine (Ala, A), a non-phosphorylatable residue (He et al, 2010; Kassardjian et al, 2012). Then, we tested the consequences for the transcriptional activity of mutant YY1 using the dual-luciferase reporter assay in ES cells. Loss of Ser247 phosphorylation resulted in a significant decrease in the ability of YY1 to transactivate luciferase reporters driven by the *Xist* promoter. Similarly, Ser→Ala substitution at position 120 also showed a trend, albeit not statistically significant, towards decline (Fig. 3L). Conversely, substitution of either Ser 120 or 247 with aspartic acid (Asp, D), which was used to mimic the phosphorylated state (Kassardjian et al, 2012; Li et al, 2004; Riman et al, 2012; Trautwein et al, 1993), showed a comparable transcriptional activity with WT YY1. These results were also confirmed in 293T cells (Appendix Fig. S6D). Finally, we overexpressed exogenous wild-type or mutant *Yy1* in differentiating ES cells, either in the presence of endogenous YY1 (wild-type) or in a YY1-degron background in which endogenous YY1 is depleted (Appendix Fig. S6A–C), and analyzed the effects on endogenous *Xist* expression. Non-phosphorylatable Ser→Ala substitutions at Ser247 reduced *Xist* expression in both wild-type and YY1-depleted cells, indicating that phosphorylation at this site is required for full YY1's transcriptional activity. By contrast, although it showed no significant effect in wild-type cells (Appendix Fig. S6E), a phospho-mimic Ser→Asp mutant at Ser247 significantly enhanced *Xist* expression in the YY1-degron background (Fig. 3M). Our findings indicated that phosphorylation of YY1 is an important mechanism downstream of FGF4–ERK signaling that regulates *Xist* upregulation.

## FGF4 is critical for the timely decline of pluripotency factors that are major repressors of Xist upregulation

*Xist* upregulation and initiation of random XCI are tightly coupled with the exit from pluripotency and the entry into differentiation. Since our results (Appendix Fig. S7A), together with a previous study (Kunath et al, 2007), indicated the absence of *Fgf4* impeded ES cell differentiation. In agreement with the impeded differentiation, both *Fgf4* deficiency and FGFR inhibition resulted in a delayed decline of core pluripotency factors (*Nanog*, *Prdm14*, and *Rex1*) during ES cell differentiation (Fig. 4A–C). We asked if impeded differentiation contributes to random XCI failure in *Fgf4* KO-84# ES cells. To this end, we used retinoic acid (RA) to facilitate the differentiation of *Fgf4* KO-84# ES cells, as revealed by both increased differentiation markers and accelerated decline of pluripotency factors comparable to that of the wild-type ES cells (Appendix Fig. S7B, C). Unexpectedly, although RA-induced ES cell differentiation has been well-established to drive random XCI (Makhlouf et al, 2014; Navarro et al, 2008; Wutz and Jaenisch, 2000), neither *Xist* upregulation and coating nor XCI state could be rescued by RA-induced differentiation. Importantly, the deficiency of both *Xist* upregulation and XCI state in RA-induced *Fgf4* KO-84# ES cells was rescued by exogenous FGF4 addition, and further blocked by FGFR inhibition (Fig. 4D–F). These results indicated that although random XCI initiation was thought to depend on differentiation, enforced induction of differentiation per se, is not enough to rescue *Fgf4* KO-84# cells from XCI failure.

It is worth mentioning that several pluripotency factors showed a transient upregulation in *Fgf4* KO or FGFR-inhibited WT cells during the initial differentiation, probably due to the effect of FGF signaling inhibition in prompting the ground state of pluripotency (Ficz et al, 2013). In contrast, *Sox2* and *Oct4* did not show notable changes due to

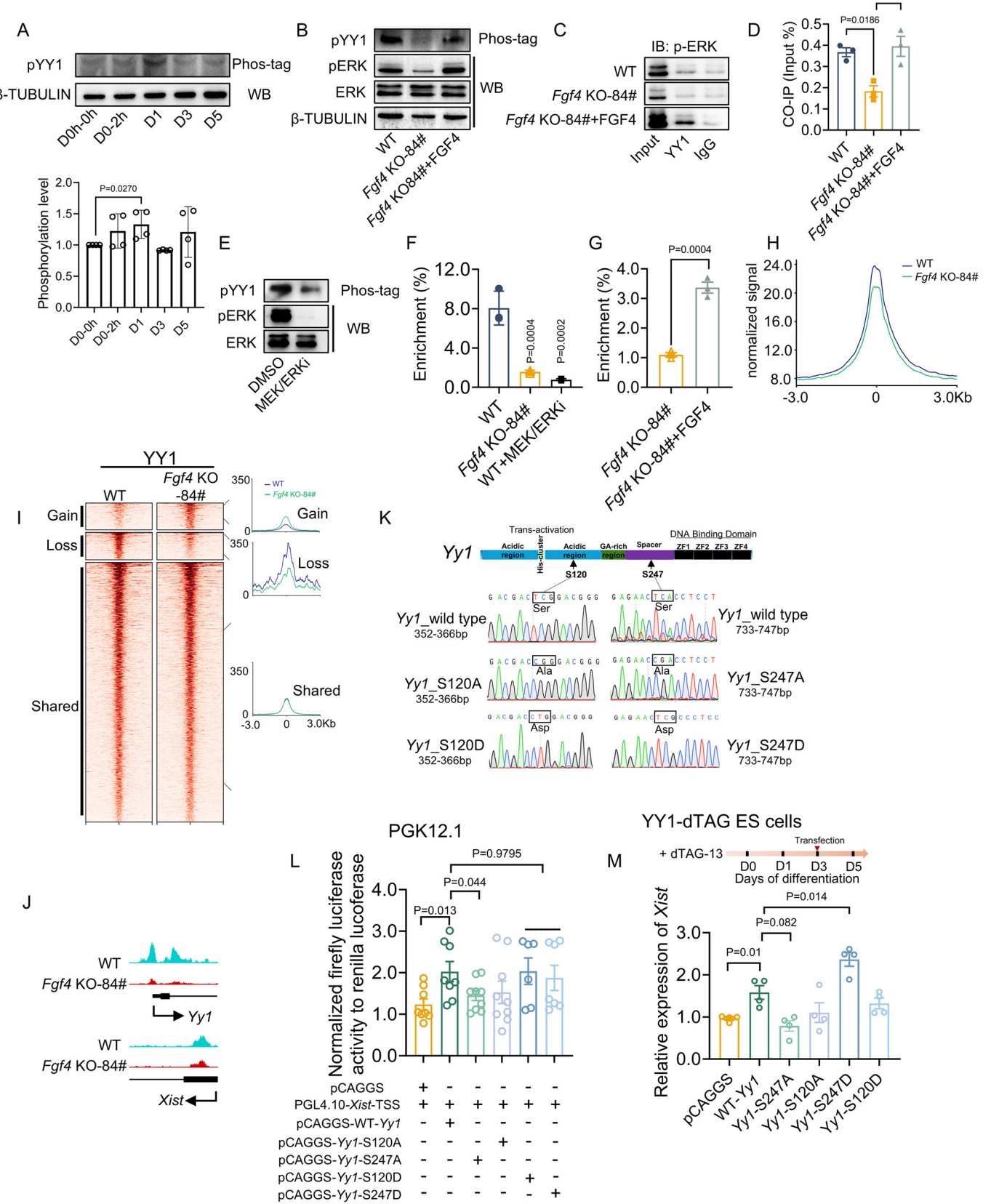

**Figure 3.   FGF4 regulates *Xist* upregulation via the ERK–YY1 axis.**

(A) Phos-tag analysis of pYY1 and Western blot analysis of β-TUBULIN in WT ES cells on day 0–0 h, day 0–2 h, day 1, day 3, and day 5 of differentiation (upper panel), along with the quantitative analysis of pYY1 phosphorylation levels (down panel), $n = 4$. (B) Phos-tag analysis of pYY1 and Western blot analysis of pERK, ERK in WT, *Fgf4* KO-84#, *Fgf4* KO-84# + FGF4 ES cells on day 0–2 h of differentiation; FGF4 10 ng/mL. (C, D) Co-IP assays (C) and quantification (D) of interaction between YY1 and pERK, in differentiating WT, *Fgf4* KO-84#, and *Fgf4* KO-84# + FGF4 ES cells on day 1 of differentiation; $n = 3$. (E) Western blot analysis of pYY1, pERK, ERK in WT treated with or without MEK/ERKi on day 0–2 h of differentiation. MEK/ERKi, PD0325901, 1 μM. (F) ChIP analysis of YY1 enrichment at the binding site within the *Xist*'s 5′ region in WT, *Fgf4* KO-84#, and WT ES cells treated with MEK/ERKi on day 1 of differentiation; $n = 3$. (G) ChIP analysis of YY1 enrichment at the binding site within the *Xist*'s 5′ region in *Fgf4* KO-84# after FGF4 addition on day 1 of differentiation; $n = 3$. (H) Average profile (metaplot) of normalized YY1 CUT&Tag signal centered on transcription start sites in WT and *Fgf4* KO-84# ES cells. Signals were plotted across a ± 3 kb window around the TSS. The x-axis indicates distance from the TSS (kb), and the y-axis indicates normalized YY1 enrichment. (I) Heatmaps of YY1 binding enrichment and corresponding average CUT&Tag signal profiles in WT and *Fgf4* KO-84# ES cells. CUT&Tag signals are plotted around TSS ( ± 3 kb) associated with YY1 peaks. YY1 peaks are categorized as gained ($n = 187$), lost ($n = 191$), or shared ($n = 1809$) in *Fgf4* KO-84# compared with WT. (J) The UCSC browser view showing YY1 enrichment at representative promoter regions in WT, *Fgf4* KO-84# ES cells. Gene models are shown below; arrows indicate the direction of transcription. (K) Schematic diagram of DNA mutations via the substitution of serine (Ser, S) 120 and 247 with alanine (Ala, A, non-phosphorylation-mimic mutation) or aspartic acid (Asp, D, phosphorylation-mimic mutation) with the coding region of *Yy1* gene. For the Ser→Ala substitution at the positions 120 or 247, the triplet codons TCG and TCA were respectively replaced by CGG and CGA; For the Ser→Asp substitution at the position 120 or 247, the triplet codon TCG and TCA were replaced by CTG and TCG, respectively. (L) Relative luciferase activity of the reporters containing *Xist* promoter, which were co-transfected with an empty vector (pCAGGS), or overexpression vector of WT or mutant *Yy1* in WT ES cells; $n = 6$–9. Each dot indicates the result of each reaction, and assays were replicated independently at least three times. (M) Relative expression levels of endogenous *Xist* in YY1-dTAG ES cells transfected with an empty vector (pCAGGS), or an overexpression vector of WT or mutant *Yy1*; $n = 4$. (D, F, G, L, M) Data are shown as means ± SEM. $P$ value was calculated by one-way ANOVA test with multiple comparisons. For more details, please see the "Methods". Source data are available online for this figure.

either *Fgf4* deficiency or FGFR inhibition (Appendix Fig. S7D). Because *Nanog*, *Prdm14*, and *Rex1* have been reported as the major repressors of *Xist*, we next tested if the knockout of these pluripotency factors could rescue the random XCI state in *Fgf4* KO-84# ES cells. Our results showed that knockout of *Prdm14*, rather than *Nanog and Rex1*, partially rescued the random XCI state (Fig. 4G,H). These results suggested that FGF4 is responsible for the timely decline of several core pluripotency factors that are major repressors of *Xist* upregulation. The delayed decline of *Prdm14* may contribute to random XCI failure in *Fgf4* KO-84# ES cells.

## Discussion

A variety of positive and negative factors ensuring *Xist* upregulation have been described: transcription factors YY1 and GATA can directly drive robust *Xist* transcriptional activation (Makhlouf et al, 2014; Ravid et al, 2023); RNF12, an E3-ubiquitin ligase, can target and degrade REX1, a critical repressor of *Xist*, and acts as a dose-dependent activator of *Xist* upregulation (Jonkers et al, 2009). In contrast, the pluripotency factors such as NANOG, OCT4, and REX1 can directly repress *Xist* transcription in the pluripotent cells. In addition to the transcriptional regulators and their partners, noncoding RNAs within an X-linked region encompassing *Xist* (known as the X-Inactivation Center, Xic), including the repressive antisense transcript *Tsix* and the positive molecular switches *Jpx* and *Ftx*, etc., also play central roles in regulating *Xist* transcription (Navarro et al, 2008; Tian et al, 2010). Despite the complex and coordinated regulatory network, however, little is known regarding the signals that are layered on top of both known positive and negative regulators to trigger *Xist* expression and synchronize XCI and differentiation.

Our data demonstrated that FGF4 plays an essential role in upregulating *Xist* expression and initiating random XCI. More importantly, we found FGF4 serves as a key coordinator that is layered on top of the complex regulatory network of XCI. Although XO ES cells that had lost one X chromosome do not initiate XCI (Monkhorst et al, 2008), our data indicate that the impeded upregulation and impaired random XCI are not attributable to X chromosome loss or the health status of *Fgf4* KO-84# ES cells, in line with previous observations that FGF4 is essential for regulating differentiation commitment, but not indispensable for the propagation and survival of ES cells (Kunath et al, 2007).

Two parallel but opposing pathways are involved in FGF4-induced random XCI initiation, i.e., phosphorylation of YY1 and decline of pluripotency factors. YY1 is the first autosomal transcriptional factor that directly triggers robust *Xist* transcriptional activation at the onset of XCI(Makhlouf et al, 2014). An earlier study also reported that YY1, as an adapter between regulatory RNA and chromatin targets, tethers *Xist* RNA to the inactive X nucleation center (Jeon and Lee, 2011). Interestingly, although YY1 maintained steady-state protein levels from undifferentiated to differentiated ES cells, YY1 had bound the *Xist* locus in undifferentiated ES before XCI initiation(Makhlouf et al, 2014). Our findings not only give an explanation for this discrepancy but also highlight the presence of the FGF4–ERK–YY1 axis and its role in upregulating *Xist* and initiating random XCI.

In addition, the phosphorylation sites analyzed in this study have previously been implicated in regulating YY1 transcriptional activity in diverse cell types and processes (He et al, 2010; He et al, 2010; Kassardjian et al, 2012; Martinez-Moreno et al, 2017), and our mutational analyses extend this concept to *Xist* regulation during random XCI. Furthermore, comparative analysis of female PGK12.1 and male R1 ES cells indicates that despite total YY1 protein levels maintain at steady-state levels in both male and female ES cells, YY1 phosphorylation is more prone to occur in female cells during XCI initiation. Given that YY1 phosphorylation in our system largely depends on FGF4–ERK signaling and that ERK signaling is well known to regulate transcription factor function through phosphorylation, it is plausible that differences in upstream signaling, including ERK pathway output, contribute to the distinct YY1 phosphorylation patterns observed between XX and XY ES cells, although direct comparisons of ERK activation between PGK12.1 and R1 remain to be determined.

It is noteworthy that a temporal gap exists between ERK activation and YY1 phosphorylation. It can be speculated that upon LIF withdrawal, FGF-ERK signaling is activated, which in turn

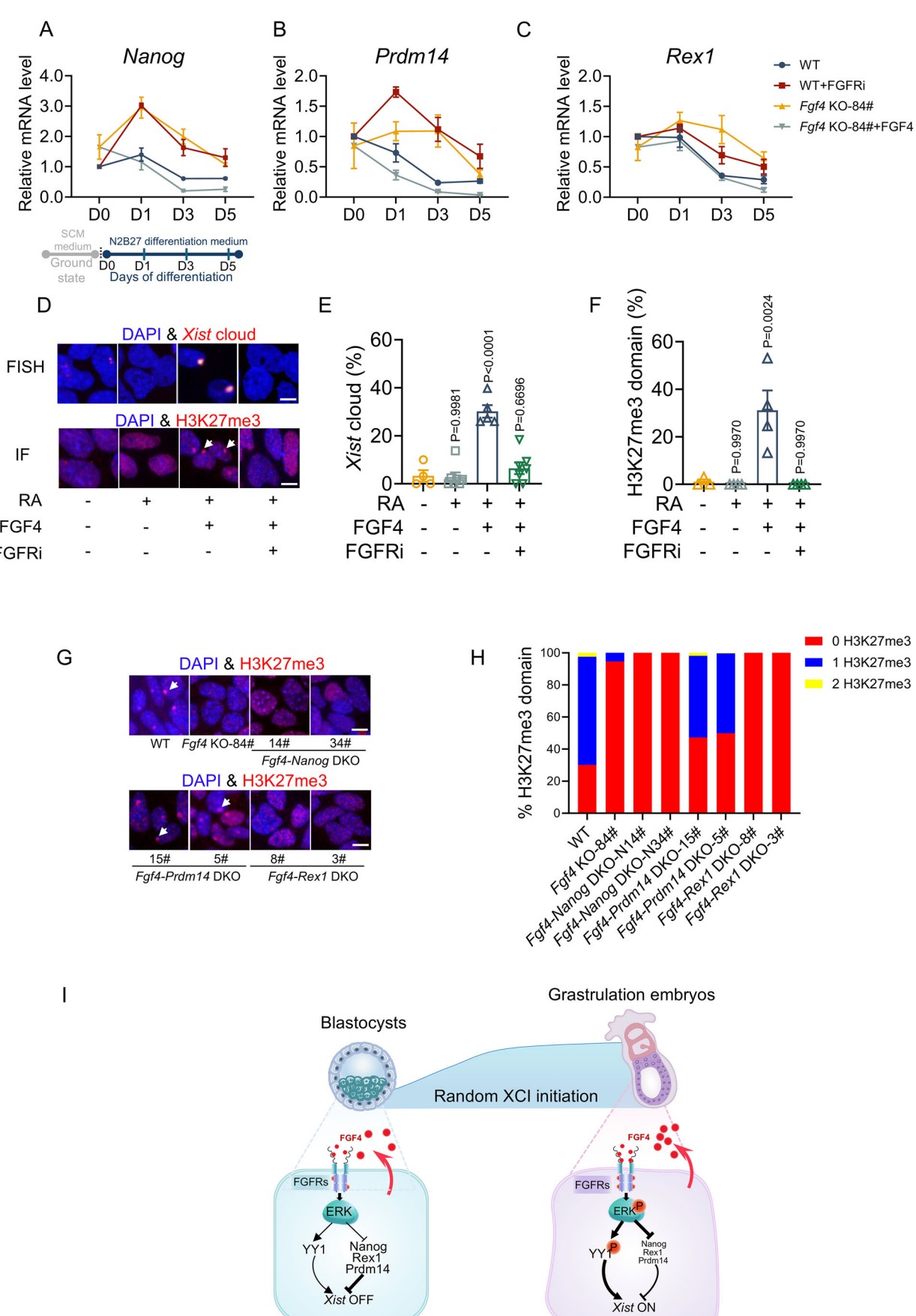

**Figure 4.  FGF4 is critical for the timely decline of pluripotency factors that are major repressors of *Xist* upregulation.**

(A–C) Dynamic expression levels of *Nanog* (D), *Prdm14* (E), *Rex1* (F) in WT, WT+FGFRi, *Fgf4* KO-84#, *Fgf4* KO-84# + FGF4 ES cells at days 0, 1, 3, and 5 of differentiation; n = 3. The schematic diagram below the X-axis in D indicates the timeline of differentiation. The day when ES cells were transferred to the differentiation medium was designated as day 0. (D–F)*Xist* FISH staining and immunofluorescence staining of H3K27me3, and corresponding quantification of *Xist* clouds (B) and H3K27me3 domains (C) in *Fgf4* KO-84 ES cells treated with retinoic acid (RA), FGF4, and pan-FGFR inhibitor, either alone or in combination; RA, 100 nM; FGF4, 10 ng/mL; FGFRi: pan-FGFR inhibitor (BGJ398), 75 nM; n = 4–7 for (B); n = 3–4 for (C); scale bars: 10 μm. In the upper panels, the white arrow in the upper panels denotes a nuclear *Xist* cloud. % *Xist*-cloud = number of cells with *Xist* cloud/total number of analyzed cells. Cells with one *Xist* cloud signal were considered as normal XCI and included in the statistical unit. In the lower panels, the white in the nuclear H3K27me3 domain. % H3K27me3 = number of cells with nuclear H3K27me3 domains/total number of analyzed cells. Cells with one H3K27me3 domain were considered normal XCI and included in the statistical unit. (G, H) Immunofluorescence staining (G) and quantification (H) of H3K27me3 domains in WT, *Fgf4* KO, *Fgf4-Nanog* DKO, *Fgf4-Prdm14* DKO, *Fgf4-Rex1* DKO ES cells on day 5 of differentiation; scale bars: 10 μm. (I) A model illustrating the mechanism underlying FGF4-induced *Xist* upregulation and XCI initiation. The white arrow denotes a nuclear H3K27me3 domain. % H3K27me3 = number of cells with nuclear H3K27me3 domains/total number of analyzed cells. Cells with one H3K27me3 domain were considered as normal XCI and included in the statistical unit. (B, C) Data are shown as means ± SEM. *P* value was calculated by one-way ANOVA test with multiple comparisons. For more details, please see the "Methods". Source data are available online for this figure.

initiates the transition from a pluripotent state toward differentiation (Liu et al, 2013). Subsequently, the pre-existing YY1 is phosphorylated under the induction of ERK, thereby activating its transcriptional activity (Li et al, 2012). It is also worth mentioning that YY1 phosphorylation is not a simple binary on/off switch but is instead finely tuned by the signaling context. We do not rule out the possibility that other mechanisms or factors mediate the effects of FGF-ERK signaling on *Xist* upregulation and XCI initiation, since the changes induced by the YY1 mutation are smaller than those caused by either *Fgf4* knockout or ERK inhibition. Our data also suggest that, despite the loss of YY1 binding at many sites due to FGF4 knockout, YY1 binding at a large number of sites does not seem to rely on FGF4.

A previous study by Schulz et al provided a mechanistic link between the X chromosome dose and developmental progression by elucidating that two active X chromosomes stabilize the naïve pluripotent state by inhibiting the ERK pathway, thus delaying ES differentiation (Schulz et al, 2014). By contrast, our results identify that once exiting the pluripotent state, ERK, in turn, triggers random XCI. Thus, Schulz's results and our findings support the reciprocal antagonism of ERK and X chromosome dose. ERK is a prominent pathway engaged by FGF signaling (Brewer et al, 2016), and its activation was thought to repress pluripotency factors (Ficz et al, 2013; Schrode et al, 2014), the direct and major repressor of *Xist* expression (Navarro et al, 2008). We have shown that *Fgf4* deficiency delayed the timely decline of pluripotency factors such as *Nanog*, *Prdm14*, and *Rex1*, all of which have been reported as the regulatory targets of ERK (Chen et al, 2015). Among them, *Prdm14* is a crucial regulator of mouse primordial germ cells epigenetic reprogramming and pluripotency. In previous studies, *Prdm14* displayed a potential function in repressing *Xist* in ES cells (Ma et al, 2011), and its overexpression accelerates X-chromosome reactivation during the conversion of epiblast stem cells to ES cells (Gillich et al, 2012). Our data suggested that *Prdm14* may be an important downstream mediator that facilitates the inductive effect of FGF4 in initiating random XCI, because knockout of *Prdm14*, but not other pluripotency factors, partially rescued random XCI. It should also be mentioned that although RA can accelerate the decline of *Prdm14* in *Fgf*$^{-/-}$ ES cells, it still cannot rescue XCI, suggesting that the effect of pluripotency factor decline on random XCI depression may be dose-dependent.

Our present study also provides supporting evidence of autonomous XCI. Although XCI is thought to be cell autonomous (Del et al, 2017; Pacini et al, 2021), whether and how autocrine

signaling regulates XCI remains unclear. Previous studies, mainly focused on transcriptional regulators and their partners, as well as noncoding RNAs within Xic, although paracrine Hedgehog by the surrounding visceral endoderm has been identified as the critical factor for random XCI initiation (Del et al, 2017), whether this process depends on the autocrine growth factor produced within the epiblast or ES cells, remains to be determined. We took advantage of the ES cell model that mimics random XCI initiation upon differentiation. This model helps eliminate potential compensation and interference from maternal factors, thus having an advantage in studying cell-autonomous mechanisms (Del et al, 2017; Pacini et al, 2021). FGF4 is the prominent autoinductive stimulus for ES cells to exit the pluripotent state and peri-implantation embryogenesis progresses (Feldman et al, 1995; Kunath et al, 2007; Molotkov et al, 2017). Our results indicated that either blocking of FGFR signaling or FGF4 deficiency led to nearly complete failure of *Xist* upregulation and H3K27me3 deposition, demonstrating that FGF4 is the essential autocrine signal that triggers random XCI.

In summary, we identify that autocrine FGF4 plays a crucial role in synchronizing the initiation of random XCI with differentiation, via the comprehensive regulation of the opposing pathways (Fig. 4I). The FGF4–ERK-YY1 axis not only updates the current knowledge about how autocrine factor and intracellular signaling ensure *Xist* transcriptional activation and onset of XCI, but also provides direct evidence of how an autocrine signal produced within the epiblast and ES cells triggers a developmental epigenetic hallmark event.

## Methods

**Reagents and tools table**

| Reagent/resource | Reference or source | Identifier or catalog number |
|---|---|---|
| **Experimental models** | | |
| ICR mice | SPF Biotechnology Co., Ltd (Beijing, China) | N/A |
| PGK12.1 | Gift from Neil Brockdorff, University of Oxford and Ingolf Bach, University of Massachusetts Medical School | N/A |

| Reagent/resource | Reference or source | Identifier or catalog number |
|---|---|---|
| **Recombinant DNA** | | |
| pCAGGS-oe-*Yy1* | This study | N/A |
| pCAGGS-*Yy1*-S247A | This study | N/A |
| pCAGGS-*Yy1*-S120A | This study | N/A |
| pCAGGS-*Yy1*-S247D | This study | N/A |
| pCAGGS-*Yy1*-S120D | This study | N/A |
| pGL4.10-*Xist*-promoter | Makhlouf et al, 2014 | N/A |
| pRL-TK | Promega | Cat #E2241 |
| pX330.puro | Gift from Xie lab, Tsinghua University | N/A |
| pMK289-FKBP$^{F36V}$-GFP | Gift from Xie lab, Tsinghua University | N/A |
| **Antibodies** | | |
| Human FGF-4 antibody | R&D System | Cat# AF235 |
| FGF Receptor 2 (D4L2V) Rabbit mAb | Cell Signaling Technology | Cat# 23328 |
| Phospho-FGF Receptor (Tyr653/654) | Cell Signaling Technology | Cat# 3471 |
| p44/42 MAPK (Erk1/2) Antibody | Cell Signaling Technology | Cat# 9102 |
| Phospho-p44/42 MAPK (Erk1/2) | Cell Signaling Technology | Cat# 4376 |
| Mouse/Rat Nestin MAb (Clone 307501) antibody | R&D System | Cat# MAB2736 |
| Nanog (D2A3) XP Rabbit mAb (Mouse Specific) antibody | Cell Signaling Technology | Cat# 8822 |
| YY1 (H-10) antibody | Santa Cruz | Cat# sc-7341 |
| Protein A/G PLUS-Agarose antibody | Santa Cruz | Cat# sc-2003 |
| Rabbit Anti-Histone H3, trimethyl (Lys27) Polyclonal antibody | Millipore | Cat# 07-449 |
| Goat Anti-Mouse IgG (H + L) Highly Cross-adsorbed Antibody, Alexa Fluor 488 | Invitrogen | Cat# A-11029 |
| Goat anti-Rabbit IgG (H + L) Cross-Adsorbed Secondary Antibody, Alexa Fluor 594 | Invitrogen | Cat# A-11012 |
| Horseradish enzyme-labeled goat anti-mouse IgG (H + L) antibody | ZSGB-Bio | Cat# ZB-2305 |
| Horseradish enzyme-labeled goat anti-rabbit IgG (H + L) antibody | ZSGB-Bio | Cat# ZB-2301 |
| YY1 (D5D9Z) Rabbit Monoclonal Antibody | Cell Signaling Technology | Cat# 46395 |
| Hyperactive pA/G-MNase CUT&RUN Assay Kit for PCR/qPCR | Vazyme | Cat# HD103-02 |
| TransNGS® CUT&Tag Library Prep Kit for Illumina® | TransGen Biotech | Cat# KP172-01 |

| Reagent/resource | Reference or source | Identifier or catalog number |
|---|---|---|
| **Oligonucleotides and other sequence-based reagents** | | |
| PCR primers | This study | Appendix Table S1 |
| ChIP-qPCR primers | This study | Appendix Table S1 |
| **Chemicals, enzymes, and other reagents** | | |
| N2 supplement | Gibco | Cat# 17502048 |
| B27 supplement | Gibco | Cat# 17504044 |
| Neurobasal medium | Gibco | Cat# 21103049 |
| DMEM/F12 | Gibco | Cat# 11320-033 |
| Knockout DMEM | Gibco | Cat# 10829-018 |
| Albumin from bovine serum | Sigma-Aldrich | Cat# A1933 |
| Triton-X-100 | Sigma-Aldrich | Cat# V900502 |
| leukemia inhibitory factor (LIF) | Millipore | Cat# ESG1107 |
| 0.1% Gelatin Solution | Millipore | Cat# ES-006-B |
| GlutaMA Supplement | Gibco | Cat# 35050-061 |
| Fetal Bovine Serum | Hyclone | Cat# SH30070.03 |
| EmbryoMax MEM Non-Essential Amino Acids | Millipore | Cat# TMS-001-C |
| Penicillin–Streptomycin (P/S) | Gibco | Cat# 15140-122 |
| β-mercaptoethanol | Gibco | Cat# 21985-023 |
| Trypsin-EDTA | Gibco | Cat# 25300-054 |
| DPBS | Gibco | Cat# C14190500BT |
| RIPA Lysis Buffer | Beyotime | Cat# P0013B |
| Protease and phosphatase inhibitor cocktail for general use | Beyotime | Cat# P1045 |
| Pageruler TM prestained protein ladder | Thermo Fisher | Cat# 26616 |
| Trizol Reagent | Ambion | Cat# 15596018 |
| Direct to PCR (cell) | VIAGEN | Cat# 301-C |
| *Xist* probe | Invitrogen | Cat# VX-01 |
| Hoescht | Vectashield | Cat# H1200 |
| Lipofectamine 2000 Transfection Reagent | Invitrogen | Cat# 11668019 |
| Cells-to-cDNA II Cell Lysis Buffer | Thermo Fisher | Cat# AM8723 |

| Reagent/resource | Reference or source | Identifier or catalog number |
|---|---|---|
| Recombinant Mouse FGF-4 Protein | R&D System | Cat# 5846-F4-025 |
| Fibronectin Human Protein, Native | Thermo Fisher | Cat# PHE0023 |
| Retinoic acid | Sigma-Aldrich | Cat# R2625-100MG |
| FGFRi (BGJ398) | Selleck | Cat# S2183 |
| GSK-3i (CHIR-99021) | Selleck | Cat# S2924 |
| MEKi (PD0325901) | Selleck | Cat# S1036 |
| Fludarabine | Selleck | Cat# S1491 |
| SH-4-54 | Selleck | Cat# S7337 |
| U73122 | Selleck | Cat# S7791 |
| Phorbol 12-myristate 13-acetate | Selleck | Cat# S8012 |
| LY2109761 | Selleck | Cat# S2704 |
| Colchicine | Beyotime | Cat# ST1173-250mg |
| NP-40 Lysis Buffer | Beyotime | Cat# P10013F |
| Phosbind Acrylamide | APExBIO | Cat# F4002 |
| SsoFast EvaGreen Supermix | Bio-rad | Cat# 172-5201 |
| ChromaFlash High-Sensitivity ChIP Kit | EpiGentek | Cat# P-2027 |
| PrimeSTAR HS DNA Polymerase | Takara | Cat# R044 |
| HiScript II Q Select RT SuperMix for qPCR | Vazyme | Cat# R233-01 |
| ViewRNA ISH Cell Assay Kit | Invitrogen | Cat# QVC0001 |
| BCA protein quantification kit | Beyotime | |
| eECL Western Blot Kit | CWBIO | Cat# CW0049S |
| Dual-Luciferase Reporter Assay System | Promega | Cat# E1910 |
| Annexin V-FITC Apoptosis Detection Kit | Beyotime | Cat# C1062M |
| **Software** | | |
| ImageJ | Schneider et al, 2012 | https://imagej.nih.gov/ij/ |
| GraphPad Prism 8 | GraphPad Software Inc. | https://www.graphpad.com/ |
| **Other** | | |
| Hydrophobic PVDF membranes | Amersham | Cat# 10600023 |
| Fisherbrand microscope cover glass | Fisher scientific | Cat# 12-545-101 |

## Mouse embryo preparation and collection

All experimental procedures were approved by and performed in accordance with the guidelines of the Institutional Animal Care and Use Committee of China Agricultural University (No. AW31305202-1-1). ICR mice used in the present study were provided by SPF Biotechnology Co., Ltd (Beijing, China). Eight-week-old female mice of spontaneous estrus were mated with 10-week-old ICR male mice. The morning after mating, copulatory plugs were monitored. Pregnant mice received daily intraperitoneal injections of FGFR inhibitor infigratinib, MEK/ERK inhibitor PD0325901, MEK/ERK agonist Phorbol 12-myristate 13-acetate from E5.0 to E7.5. For post-implantation female embryo collection, E6.5 or E7.5 conceptuses covered with the decidual mass were gently teased away from the uterus. The epiblasts were separated from the extraembryonic ectoderm and rinsed twice in 0.1% DPBS-PVA before further analyses. The ectoplacental cones were used for sex determination by PCR.

## Mouse ES cells culture and differentiation

Mouse embryonic stem cells (ESCs) used in this study were *PGK12.1* ES cells, kindly provided by Prof. Neil Brockdorff, and maintained under standard culture conditions as previously described. ES cells were cultured in stem cell medium (SCM) containing Knockout DMEM base medium, 10% fetal bovine serum, 1× non-essential amino acids, 2 mM GlutaMAX, 55 μM β-mercaptoethanol, $1 \times 10^5$ units leukemia inhibitory factor (LIF), and 1% penicillin–streptomycin, and incubated in a humidifier containing 5% $CO_2$ at 37 °C. To induce differentiation, $3.0–5.0 \times 10^3/cm^2$ cells were planted onto glasses coated with 5 μg/cm² fibronectin in SCM. After 24 h growth, cells were rinsed by 1 ml DPBS and replaced with differentiation medium, (N2B27 medium, 5 mg/mL BSA, 1% penicillin–streptomycin). Subsequently, the medium was changed daily. For *Fgf4* KO-84# rescue group, add 10 ng/mL FGF4 recombinant protein and 1 μg/ml heparin in the differentiation culture medium while culturing the cells.

## Reconstruction of the spatiotemporal expression pattern of FGF ligands and receptors

The spatiotemporal expression pattern of FGF ligands and receptors was reconstructed using the online tool (eGastrulation: http://egastrulation.sibcb.ac.cn), a 3D gene expression database of gastrulating mouse embryos (Peng et al, 2016; Peng et al, 2019). The current version of the database is built from expression data collated from the embryo which was serially cryo-sectioned and samples from different regions (ectoderm, mesoderm, and endoderm). On each section, ~20 cells were isolated by laser capture microdissection from the ectoderm, mesoderm, and endoderm in each of the quadrants: anterior (A), posterior (P), and lateral (left/right, L/R), of each sample section (S1, S2, S3… etc, with section 1 being the most distal). The position of each cell sample and expression levels of FGF ligands and receptors were presented by the average color intensity and plotted in the corn plot format (so named for its resemblance to a corn cob).

## Quantitative real-time polymerase chain reaction (qRT-PCR)

Embryos or cells used for RNA extraction were washed with DPBS twice and transferred into a 1.5-ml centrifuge tube, and then, 1 mL TRIzol reagent was added. After 10 min of incubation on ice, samples were cryopreserved into ultra-low temperature freezer till use. Total RNA was extracted from embryos or cells following the manufacturer's instructions with TRIzol reagent. HiScript II Q Select RT SuperMix for qPCR kit was used to reverse-transcribe 1 μg total RNA into cDNA according to the manufacturer's protocol. Quantitative real-time PCR analysis was performed using SsoFast EvaGreen Supermix. Primers used in the present study are shown in Table S1. Gene relative expression was referenced to Gapdh and calculated by the $2^{-\Delta\Delta Ct}$ method.

## Western blot

Embryos samples were used to detect protein level were rinsed by 0.1% DPBS-PVA twice, and collected into centrifuge tube on ice, and 10 μL 2×lammli sample buffer (Bio-rad) with 5% β-mercaptoethanol was added following 100 °C boiled for 10 min.

For ES cells, ice-cold DPBS buffer was used to wash cells three times. Cells were then lysed using either RIPA buffer or 40% NP40 lysis buffer (for Phos-tag analysis) containing 1× protease cocktail, and cells were suspended and shaken several times on ice. Lysis solution was then centrifuged at $12,500 \times g$ for 10 min, and the total concentration of supernatant protein was measured by the BCA protein quantification kit. Proteins were then extracted and separated using an 12% acrylamide gel with or without Phos-tag acrylamide ('+ Phos-tag' or '- Phos-tag', respectively). If for Phos-tag, before transferred, the gel needs to be washed three times with 10 mM EDTA transfer buffer, followed by one wash with transfer buffer to remove residual metal ions. The proteins were then transferred to a PVDF membrane following 5% skim milk blocking for 1 h at room temperature. Primary antibody p44/42 MAPK (Erk1/2) (1:1000), Phospho-p44/42 MAPK (Erk1/2) (1:1000), FGF Receptor 2 (D4L2V) Rabbit mAb (1:1000), Phospho-FGF Receptor (Tyr653/654) (1:1000), YY1 (H-10) (1:1000), FGF4(1:1000) and β-tubulin (1:1000) were diluted by blocking solution and the membranes were incubated overnight at 4 °C. Then, membranes were rinsed with TBST buffer three times and incubated with secondary antibodies conjugated with HRP for 1 h at room temperature. After three washes with TBST buffer, target protein bands were visualized using eECL Western Blot Kit and detected by 5200 Imaging system (Tanon, Shanghai, China).

## Immunofluorescence analysis

In all, 4% paraformaldehyde in DPBS was used to fix embryos (overnight at 4 °C) or cells (20 min at room temperature) after the culture medium was removed. After three washes by DPBS, samples were permeabilized with 0.5% Triton-X 100 for 0.5 ~ 1 h at room temperature, and blocked with 1% BSA-PBS at 4 °C for 2–6 h. Next, diluted primary antibodies Nestin (1:200), NANOG (1:1000), or H3K27me3 (1:1000) were used to incubate with samples overnight at 4 °C. After three washes by DPBS, labeled secondary antibodies Alexa Fluor-488 (1:1000) or Alexa Fluor-594 (1:1000) were respectively added into samples under a dark environment for

1 h at room temperature. Samples were then counterstained with DAPI, and imaged using an BX51 microscope (Olympus, Tokyo, Japan) accompanied with digital microscope camera (Olympus). All photographs were quantitated by ImageJ software (Rawak Software Inc., Stuttgart, Germany).

## Chromatin immunoprecipitation (ChIP) assay

Differentiated cells were cross-linked with 1% formaldehyde and subjected to ChIP assay according to the protocols of ChromaFlash High-sensitivity ChIP kit. The DNA lysate was crushed by ultrasonic breaker (Covarias) to produce 100–700 bps fragments. To obtain input controls, 1 μg DNA in each group was released and purified, and diluted to ten times before use. Primary antibody YY1 and negative control non-immune IgG were used to precipitate binding DNA fragments. Quantitative PCR was performed, and enrichment was calculated as follows: % Input = 100 × 2(Ct (adjusted input) − Ct (IP)), Ct (adjusted input) = input Ct − 3.32. ChIP-qPCR primers are shown in Table S1.

## RNA fluorescence in situ hybridization (FISH)

*Xist* FISH was performed according to the manufacturer's instructions and our previous publication (PMID: 33061820) (Tan et al, 2016). Briefly, fixed cells were pretreated with Detergent Solution QC and digested with proteinase K, then incubated with *Xist* Probe set for 3 h at 40 °C. After PreAmplifier and Amplifier solution, Label Probe was added for hybridization for 30 min at 40 °C. Then, the cells were incubated with DAPI for 5 min, and images were acquired by a fluorescence microscope (Olympus).

## *Fgf4* knockout

CRISPR/CAS9n was used to knock out the *Fgf4* gene in PGK12.1. Using the online guide designer Benchling, we acquired and synthesized a small guide RNA (sgRNA). sgRNAs were constructed to pSpCas9n(BB)-2A-Puro vector (PX462, purchased from Addgene), at BbsI sites. And transfected into PGK12.1 using Lipofectamine 2000 (Invitrogen) following the manufacturer's instructions. 24 h after transfection, 2 μg/mL puromycin was added to the SCM to screen positive clones. Single clones were picked out after 5 days of consistent screening, and PCR product sequencing was performed to identify knockout cell lines. All sgRNAs and PCR primers used for identification of knockout cell lines are shown in Table S1.

## Plasmids, mutagenesis, and luciferase assays

For the analysis of the *Xist* promoter, the 5'-flanking region of the mouse *Xist* gene was cloned by PCR using mouse genomic DNA as a template and specific forward and reverse primers, as decribed by (Makhlouf et al, 2014). The PCR product was then ligated into the promoterless firefly luciferase reporter vectors pGL4.10, at SacI and EcoRV sites. The mouse *Yy1* gene CDS region was cloned by PCR using mouse embryo cDNA as a specific reverse and forward primer (Table S1). A 1.245 kb PCR product was inserted into the PCAGGS vector at EcoRI and XhoI sites to obtain the *Yy1* overexpression vector pCAGGS-*Yy1*.

In point mutants of *Yy1* at serine 247 and serine 120 to Alanine (Ala, A) or Aspartic (Asp, D), PCR fragments were amplified from

pCAGGS-*Yy1* using reverse and forward primers (Table S1). Mutagenesis was using Fast MultiSite Mutagenesis System from TransGen Biotech (Beijing, China), according to the manufacturer's instructions. Primers were designed using the Transgen Primer Design Web site. All mutations were confirmed by sequencing.

ES cells were seeded into 6-well plates and normal differentiation, on the day 3 of ES cells differentiation, transfected with 1.8 μg of various reporter vectors, 2 μg of expression vectors, and 0.2 μg of pRL-TK (Promega) unless otherwise indicated using Lipofectamine 2000, according to the manufacturer's instructions. After 48 h, the cells were washed in DPBS and lysed in 1× passive lysis buffer (Promega). Aliquots of 20 μL cell lysate were used to measure luciferase activity by the addition of 100 μL Luciferase Assay Reagent, followed by luminescence quantitation in a TD-20e luminometer (Turner Designs). All experiments were carried out in triplicates, and the firefly luciferase activity was normalized by the Renilla luciferase activity.

## Flow-cytometry assay

The Annexin V-FITC Apoptosis Detection Kit utilizes FITC-labeled recombinant human Annexin V to detect phosphatidylserine (PS) exposure on the outer membrane of apoptotic cells. For the determination of apoptosis in ES cells, we used Propidium Iodide (PI) for identifying necrotic or late apoptotic cells with compromised membrane integrity. First, the ES cells were cultured in 6-well plates and induced to differentiate for 3 days. Afterward, the cells were washed and digested with trypsin 1 min at room temperature, and then used DPBS resuspending cells, incubated by Annexin V-FITC and PI at room temperature in the dark for 10-20 min. To improve staining, gently resuspend the cells 2–3 times during incubation. Finally, the cells were isolated and subjected to flow cytometry analysis to determine the ratios of the cells in different cell-cycle phases.

## Karyotype analyses

For karyotype analysis, 0.2 μg/mL colchicine was added to the ES cells culture medium, and the cells were incubated for 2–4 h. The ES cells were digested into single cells by 0.05% TrypLE™ and collected by centrifugation at 1800 rpm for 6 min. ES cells were resuspended with 0.075 M KCl hypotonic solution and incubated at 37 °C for 20 min. Then, ES cells were fixed with methanol and acetic acid at a ratio of 3:1, and this process was repeated three times, each incubation lasting 20 min at room temperature. The fixed samples were sent to Haoyu Biotechnology Co., Ltd. in Guangzhou for analysis. For each cell line, more than 30 cells at metaphase were examined.

## YY1-dTAG cell construction

The donor plasmid used to tag the endogenous mouse YY1 protein was generated by modifying a pMK289 plasmid (gift from the Xie laboratory). Two 750 bp homology arms flanking the stop codon of the *Yy1* gene were amplified by PCR from genomic DNA. The sequence encoding FKBP$^{F36V}$-GFP and the donor backbone were assembled by Gibson assembly. To target *Yy1*, an sgRNA sequence targeting the stop-codon region (CACCGTCTTCTCTC

TTCTTTTCAC) was cloned into the pX330 plasmid (Addgene #42230). For cell line generation, the donor plasmid and the sgRNA/Cas9 plasmid were co-transfected into PGK12.1 ES cells using Lipofectamine 3000 (Thermo Fisher Scientific). Twenty-four hours after transfection, 2 μg/mL puromycin was added to the SCM to screen positive clones. After 5 days of selection, single-cell clones were isolated and plated into gelatin-coated 96-well plates for expansion. Resulting clones were genotyped by junction PCR, and the fusion alleles were validated by Sanger sequencing. Homozygous Yy1–FKBP$^{F36V}$-GFP clones were further validated by western blotting. YY1–FKBP proteins were acutely depleted by adding 500 nM dTAG-13 (MCE, Cat. No. HY-114421). For time-course experiments, protein degradation was induced and samples were collected at the indicated time points.

## Chromatin profiling by CUT&RUN and CUT&Tag

Chromatin profiling was performed using CUT&Tag and CUT&RUN as previously described (Kaya-Okur et al, 2019; Skene and Henikoff, 2017) with minor modifications. All antibodies, kits, and sequencing platforms are listed in the Reagents and Tools Table.

For CUT&Tag, experiments were performed using the TransNGS® CUT&Tag Library Prep Kit for Illumina® (TransGen Biotech, Beijing, China) following the manufacturer's protocol with minor modifications. In brief, $1 \times 10^5$ ES cells per reaction were collected, washed with Cell Wash Buffer, and bound to ConA Magnetic Beads II from Box 1. After washing and resuspension in Nuclear Extraction Buffer, bead-bound cells were incubated with a primary antibody against YY1, diluted 1:50 in the kit-provided buffer, at 4 °C overnight with gentle rotation; normal IgG was used as a negative control. Cells were washed with 1× Pro-Wash Buffer and incubated with the pAG-Tn5 transposome (Box 2) diluted in Tagmentation Buffer for 1 h at room temperature. Tagmentation was initiated by the addition of Mg$^{2+}$ according to the manufacturer's instructions and carried out at 37 °C for 1 h. Reactions were stopped by adding EDTA and Lysis Enhancer, followed by Proteinase K digestion at 55 °C for 1 h to release DNA fragments, which were then purified with CUT&Tag DNA Clean Beads (TransGen) and eluted in 22 μl nuclease-free water.

For CUT&RUN, experiments were carried out using the Hyperactive pA/G-MNase CUT&RUN Assay Kit for Illumina (Vazyme, Nanjing, China) according to the manufacturer's instructions with minor adaptations. Briefly, $1 \times 10^5$ ES cells per reaction were harvested, washed with ice-cold PBS, and processed for nuclei preparation as recommended in the kit manual. Nuclei were bound to ConA magnetic beads supplied in the kit and washed in the kit wash buffer. Bead-bound nuclei were incubated with a primary antibody against YY1, diluted 1:50 in antibody buffer, at 4 °C overnight with gentle rotation; control reactions received normal IgG. After washing, nuclei were incubated with the hyperactive pA/G-MNase fusion protein for 1 h at 4 °C. Chromatin digestion was initiated by the addition of CaCl$_2$ (final 2 mM) on ice for 30 min, and cleaved fragments were released by adding the provided stop buffer and incubating at 37 °C for 10 min. Supernatants containing released DNA were collected on a magnetic rack, and protein digestion together with DNA purification were performed using the proteinase and DNA clean-up reagents included in the kit. DNA was eluted in 20 μl nuclease-free water

and used directly as a template for quantitative PCR (qPCR) analysis.

For CUT&Tag, sequencing libraries were amplified from the purified DNA using the library amplification module included in the TransNGS® kit, following the manufacturer's instructions with 10 PCR cycles. Libraries were purified and size-selected with magnetic beads to enrich for fragments of ~150–700 bp, and library quality and concentration were assessed by capillary electrophoresis (Bioanalyzer or TapeStation) and fluorometric quantification. Indexed CUT&Tag libraries were pooled equimolarly and sequenced on the Illumina NovaSeq X Plus platform (San Diego, CA, United States) in PE150 mode.

For CUT&RUN, enrichment of YY1 at selected genomic loci was quantified by SYBR Green–based qPCR using aliquots of purified CUT&RUN DNA as template. Primers used for qPCR are listed in the Appendix Table S1. Signals were normalized to input or calculated as fold enrichment over IgG control using the calculated by the $2^{-\Delta\Delta Ct}$ method.

## Statistical analysis

All experiments were replicated at least three times. Results were represented as means ± standard error of mean (SEM) and analyzed with Tests, two-tailed unpaired Student's $t$ test or ANOVA, the Student's test or correct $\chi^2$ using the SPSS version 18.0 software (IBM Corp., Armonk, NY, USA). The $P$ value below 0.05 was considered the threshold of significant statistics.

CUT&Tag data were processed by Wuhan GeneRead Biotechnology Co., Ltd. Read quality was assessed with FastQC (v0.11.9) and trimmed with Trimmomatic (v0.39; reads ≥8 nt retained). Filtered paired-end reads were aligned to mm39 using Bowtie2 (v2.4.5; --very-sensitive, max insert size 1000 bp), and PCR duplicates were removed with Picard MarkDuplicates. Peaks were called with MACS2 (v2.1.1; paired-end, $q < 0.05$) and merged across replicates into a union peak set using bedtools. Differential binding was assessed with DESeq2 (v1.18.1; default settings), defining significant regions as $P < 0.05$ and |log2 fold change | > 0.585. Peaks were annotated with ChIPseeker (v1.32.0), motif enrichment was performed with HOMER (v4.11), and normalized bigWig tracks were generated with bedGraphToBigWig and visualized in IGV (v2.8.0).

## Data availability

All data attained to support our conclusions described in this manuscript are presented in the paper and its supplementary materials. No datasets amenable to large-scale data repository deposition were generated in this study.

The source data of this paper are collected in the following database record: biostudies:S-SCDT-10_1038-S44318-026-00722-2.

## Peer review information

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

## Acknowledgements

We thank Neil Brockdorff and Ingolf Bach for generously providing ES cells line. We are also grateful to the Xie Laboratory (Tsinghua University) for sharing plasmids and degron-based methodology. We thank Dr. Yao Fu for expert guidance flow cytometry and laser confocal microscopy experiments. This work was supported by grants from the National Key Research and Development Program of China (2023YFD1300504).

## Author contributions

**Lizhu Ma**: Conceptualization; Data curation; Software; Formal analysis; Validation; Investigation; Visualization; Methodology; Writing—original draft. **Wei Fu**: Conceptualization; Data curation; Formal analysis; Validation; Investigation; Visualization; Methodology; Writing—original draft. **Lei An**: Conceptualization; Resources; Supervision; Funding acquisition; Validation; Investigation; Writing—original draft; Project administration; Writing—review and editing. **Qianying Yang**: Data curation; Methodology; Writing—original draft. **Ruiqi Hai**: Software; Formal analysis; Methodology. **Guangyin Xi**: Investigation; Methodology. **Yinjuan Wang**: Writing—review and editing. **Juan Liu**: Data curation; Methodology. **Chao Zhang**: Methodology. **Yao Fu**: Data curation; Formal analysis; Methodology. **Zhenni Zhang**: Methodology.

**Xiaodong Wang**: Conceptualization; Resources; Supervision; Funding acquisition; Validation; Investigation; Methodology; Project administration; Writing—review and editing. **Jianhui Tian**: Conceptualization; Resources; Data curation; Formal analysis; Supervision; Funding acquisition; Validation; Investigation; Project administration; Writing—review and editing.

Source data underlying figure panels in this paper may have individual authorship assigned. Where available, figure panel/source data authorship is listed in the following database record: biostudies:S-SCDT-10_1038-S44318-026-00722-2.

## Disclosure and competing interests statement

The authors declare no competing interests.

