## [Peer Review File · The EMBO Journal]

FGF4 initiates X-chromosome inactivation via activating YY1 and prompting loss of pluripotency factors

Lizhu Ma, Wei Fu, Lei An, Qianying Yang, Ruiqi Hai, Guangyin Xi, Yinjuan Wang, Juan Liu, Chao Zhang, Yao Fu, Zhenni Zhang, Xiaodong Wang, and Jianhui Tian

Corresponding author: Jianhui Tian (tianjh@cau.edu.cn)

Review Timeline:

Submission Date:	15th Jul 24
Editorial Decision:	9th Sep 24
Revision Received:	18th Mar 25
Editorial Decision:	21st May 25
Revision Received:	29th Dec 25
Accepted:	23rd Jan 26

Editor: Daniel Klimmeck

Transaction Report:

Dear Dr Tian,

Thank you again for the submission of your manuscript (EMBOJ-2024-118190) to The EMBO Journal. Please accept my sincere apologies for getting back to you with delay due to protracted referee input and detailed discussion in the editorial team. Your study was assessed by two reviewers with expertise in early developmental signaling and XCI, whose comments are enclosed below.

As you will see from the experts' reports, the referees acknowledge the analysis and potential interest of your results. However, they also express major concerns regarding completeness and in vivo relevance of the findings, which need to be addressed thoroughly to make them supportive of publication in the EMBO Journal. The reviewers also raise a number of issues related to the data presentation, additional controls and improved methods annotation required, statistics applied and overall discussion of related literature, that would need to be conclusively addressed to achieve the level of robustness and clarity needed for The EMBO Journal.

Given the overall interest stated and broader angle of your findings, we are able to invite you to revise your manuscript experimentally to address the referees' comments. I need to stress though that we do require strong support from the referees on a revised version of the study in order to move on to publication of the work.

In light of the extensive experimentation requested, I would appreciate if you could contact me during the next weeks for exchange e.g. a video call to discuss your perspective on the comments and potential plan for revisions.

Please feel free to contact me if you have any questions or need further input on the referee comments.

When submitting your revised manuscript, please carefully review the instructions below.

Please feel free to approach me any time should you have additional questions related to this.

Thank you for the opportunity to consider your work for publication.

I look forward to your revision.

Kind regards,

Daniel Klimmeck

Daniel Klimmeck, PhD
Senior Editor
The EMBO Journal

Instruction for the preparation of your revised manuscript:

- 1) a .docx formatted version of the manuscript text (including legends for main figures, EV figures and tables). Please make sure that the changes are highlighted to be clearly visible.
- 2) individual production quality figure files as .eps, .tif, .jpg (one file per figure).
- 3) a .docx formatted letter INCLUDING the reviewers' reports and your detailed point-by-point response to their comments. As part of the EMBO Press transparent editorial process, the point-by-point response is part of the Review Process File (RPF), which will be published alongside your paper.
- 4) a complete author checklist, which you can download from our author guidelines ([https://wol-prod-cdn.literatumonline.com/pb-assets/embo-site/Author Checklist%20-%20EMBO%20J-1561436015657.xlsx](https://wol-prod-cdn.literatumonline.com/pb-assets/embo-site/Author%20Checklist%20-%20EMBO%20J-1561436015657.xlsx)). Please insert information in the checklist that is

also reflected in the manuscript. The completed author checklist will also be part of the RPF.

6) It is mandatory to include a 'Data Availability' section after the Materials and Methods. Before submitting your revision, primary datasets produced in this study need to be deposited in an appropriate public database, and the accession numbers and database listed under 'Data Availability'. Please remember to provide a reviewer password if the datasets are not yet public (see <https://www.embopress.org/page/journal/14602075/authorguide#datadeposition>).

7) Our journal encourages inclusion of *data citations in the reference list* to directly cite datasets that were re-used and obtained from public databases. Data citations in the article text are distinct from normal bibliographical citations and should directly link to the database records from which the data can be accessed. In the main text, data citations are formatted as follows: "Data ref: Smith et al, 2001" or "Data ref: NCBI Sequence Read Archive PRJNA342805, 2017". In the Reference list, data citations must be labeled with "[DATASET]". A data reference must provide the database name, accession number/identifiers and a resolvable link to the landing page from which the data can be accessed at the end of the reference. Further instructions are available at .

8) At EMBO Press we ask authors to provide source data for the main and EV figures. Our source data coordinator will contact you to discuss which figure panels we would need source data for and will also provide you with helpful tips on how to upload and organize the files.

Numerical data can be provided as individual .xls or .csv files (including a tab describing the data). For 'blots' or microscopy, uncropped images should be submitted (using a zip archive or a single pdf per main figure if multiple images need to be supplied for one panel). Additional information on source data and instruction on how to label the files are available at .

9) We replaced Supplementary Information with Expanded View (EV) Figures and Tables that are collapsible/expandable online (see examples in <https://www.embopress.org/doi/10.15252/emj.201695874>). A maximum of 5 EV Figures can be typeset. EV Figures should be cited as 'Figure EV1, Figure EV2" etc. in the text and their respective legends should be included in the main text after the legends of regular figures.

11) For data quantification: please specify the name of the statistical test used to generate error bars and P values, the number (n) of independent experiments (specify technical or biological replicates) underlying each data point and the test used to calculate p-values in each figure legend. The figure legends should contain a basic description of n, P and the test applied. Graphs must include a description of the bars and the error bars (s.d., s.e.m.).

We realize that it is difficult to revise to a specific deadline. In the interest of protecting the conceptual advance provided by the work, we recommend a revision within 3 months (8th Dec 2024). Please discuss the revision progress ahead of this time with the editor if you require more time to complete the revisions.

Referee #1:

In this manuscript, the authors explore the regulatory links between the initiation of X-chromosome inactivation (XCI) and cellular differentiation; while several molecular players are known to couple pluripotency with the timings of Xist expression (e.g., Nanog, Oct4), it remains unclear which global differentiation signalling pathways are involved. Here, the authors focus on FGF4 and downstream actors and show, through a series of loss-of-function and rescue experiments, that FGF4 is upstream of XCI initiation, through MEK/ERK signalling, which leads to phosphorylation of YY1, a known activator of Xist, and timely downregulation of pluripotency factors, which repress Xist expression. Despite the general roles of FGF4, the authors succeeded in pinpointing the specific pathways that are connected to the initiation of XCI. I consider this manuscript to be very relevant and significant, extending our knowledge of which pathways are involved in the initiation of XCI. The experimental approach is solid and elegant, including appropriate controls such as rescue experiments. I have no major concerns regarding this study, and I list below my minor concerns. Importantly, the manuscript should be overall improved for clarity (both in terms of language and methodological details, which are often lacking from the results description) and for consistency (there are sometimes discrepancies in numbers of figures/panels, but also in experimental details between results and methods sections).

Minor concerns

1. The very first results that link FGF signalling to random XCI are shown in supplementary - why not in the main figure? And their description in the text lacks methodological details - can the authors be more precise about what was done?
2. The results regarding the functional window for FGF4 to initiate random XCI are very interesting. Have the authors confirmed that most cells still have two X chromosomes present after 3 days of differentiation without FGF4? If most cells have lost one X chromosome, adding FGF4 would not be expected to initiate XCI. Moreover, how healthy do the cells look after three days of differentiation without FGF4? Presumably, without XCI, XX cells would die during differentiation / not differentiate properly (Schulz et al. 2014) and/or lose an X chromosome (or XO cells are favoured). It could thus be that what the authors observed is not about a functional window for FGF4 to initiate XCI but an XX/XO issue. This alternative, if not addressed experimentally, should at least be mentioned in the text.
3. For the non-specialists, it is not clear at all why the experiments with heparin are relevant; the authors could first point out how heparin relates to FGF4.
4. Interesting that the authors observe the same effects in the rescuing experiments, whether they use FGF4 or FGF2. Is it known that FGF4 and FGF2 signal via the same receptor in mESC? Same downstream pathways? Adding explanations/clarifications to the text would be helpful.
5. How does FGF4 signalling (levels) relate to the differentiation status of mESC? Adding explanations/clarifications to the text would be helpful.
6. It would be helpful if the authors could describe in more detail the online tool they used (<http://egastrulation.sibcb.ac.cn>), particularly what the tool shows exactly. For instance, in the legend of Fig.1A, it says that "Each dot in the plot represents the cell sample at the specific positional address" - what does this mean exactly? What is the cell sample?
7. The authors show that the downregulation of pluripotency factors is impaired upon inhibition or knockout of FGF4 and show expression data at D1, D3 and D5. Why not showing expression data at D0, which is the starting point? (the figure legend actually mentions D0, but the data is not shown in the figure). The timeline of FGF4 inhibition should also be indicated, i.e., when was the inhibitor added, especially in relation to start of differentiation?
8. In general, the manuscript would benefit from including more methodological explanations along with the description of the results; I have already provided some examples. Another one would be the section on the experiments with mice, the authors could make understanding easier by including more details.

9. The letters on top of the barplots to represent statistical significance is not the most usual representation. It can be maintained if according to the journal's policies, otherwise a small table below the barplot could be used, or the more usual stars and lines specifying which comparisons were made.
10. Corrections: One of the Ser→Ala substitutions is referred to being at either position 118 (text) or position 120 (text and figure).
11. Corrections: In the Results, the mutations of YY1 are referred to as Ser→Ala substitutions, but in the Methods they are referred to Ser→Asp.
12. Figures 3H and 3I could be improved. In Fig. 3H it is confusing that the YY1 protein structure is shown below the Xist locus, and with the same length. In Fig. 3I the authors depict YY1 motifs with different shapes for wildtype and mutant YY1 forms, but the motifs are always the same, so I believe they should be represented by the same shape.
13. Corrections: Some supplementary figures are not well indicated in the main text (e.g., S1N should be S1O; S1L should be S1N; S1J-K should be S1K-L).
14. Corrections: Some figure legends have mistakes regarding the letters of the panels (e.g., Fig. 1).
15. References:
- "Initiation of random XCI strictly relies on the upregulation of long noncoding RNA Xist upon differentiation from the pluripotent state (Makhlouf et al, 2014)" This was first shown in Penny et al. 1996, not Makhlouf et al. 2014.
 - I was surprised that the authors never mentioned the work from Schulz et al. 2014, which revealed feedback regulation between MAPK signaling and initiation of X-inactivation.
 - "A variety of positive and negative factors ensuring Xist upregulation have been described: transcription factors YY1 and GATA can directly drive robust Xist transcriptional activation (Jonkers et al, 2009); RNF12, an E3-ubiquitin ligase can target and degrade REX1, a critical repressor of Xist, acts as a dose-dependent activator of Xist upregulation (Navarro et al, 2008; Tian et al, 2010)." The references seem to be mixed up. Jonkers et al. has not shown the regulation of Xist by YY1 and GATA; these findings were made by Makhlouf et al. 2014 and by Lustig et al. 2023. The role of RNF12 was instead shown by Jonkers et al. 2009.
16. The second paragraph in the discussion would have been helpful in the introduction already; it could be moved.
17. Language/writing could be improved throughout, especially for clarity.
18. Not clear what "holistically" in the title means. The syntagm "loss of pluripotency factors" is not well linked with the rest.

Referee #2:

Comments to the authors

In this manuscript, Ma et al. investigate molecular mechanisms of Xist upregulation, which triggers initiation of random XCI in female mammals. Based on analyses with mouse female ESC differentiation system, they showed that (1) FGF4 is critical for Xist upregulation upon differentiation and (2) phosphorylation of YY1 by FGF4-ERK pathway is important for direct upregulation of Xist. The authors propose that these two pathways contribute to upregulate Xist in parallel. I think their data well supports (1) although it is already accepted that FGF4 is critical for the induction of differentiation in mouse ESCs. Their data supporting (2) is, however, not strong and need to be more strengthened by additional experiments listed below. My concern is FGF4 signal may contribute simply to ESC differentiation and indirectly upregulate Xist expression, but not much contribute to YY1 phosphorylation-mediated direct upregulation of Xist.

Major points:

- First of all, there are many typos in the figure legends. For example, all statistics should be indicated with "*", not with alphabets such as "a" or "b". In addition, description in the figure legends is not enough to understand each figure. The authors should provide more information in the legend to understand each figure such as experimental conditions and explanation for abbreviations. These figure legends made the presented data not easy to understand. The authors should also provide more information in the methods. For example, it is not clear which ESC line they used in this study. Before submission of manuscript, these points have to be carefully checked and corrected.
- The authors proposed that phosphorylated form of YY1 activates Xist transcription. Is YY1 phosphorylated at which time points during ESC differentiation? A time-course experiment showing YY1 phosphorylation during differentiation should be presented like Fig. S2B&C. As YY1 has been shown to be important for sustained expression of Xist in MEFs (Makhlouf 2014), I think it is important to examine whether YY1 is phosphorylated only transiently at the beginning of differentiation or during the

whole differentiation time-windows.

3. The authors examined the importance of YY1 phosphorylation by mutating putative phosphorylation sites of YY1 and showed that S247A mutation lacks the activation of Xist-promoter-luciferase activity (Fig 3I). My concern is that even WT YY1 was able to activate luciferase activity about only 2-fold, which is quite weak. Can the authors examine the effect of YY1 mutant overexpression on endogenous Xist expression? I think this experiment is important to strengthen their hypothesis that phosphorylation of YY1 facilitates Xist expression upon differentiation. Ideally, endogenous Yy1 should be knocked out, and then exogenous Yy1 wt, S120A, and S247A will be expressed to see how much Xist can be transcribed. I also would like the authors to create the phosphorylation mimic form of YY1 mutants, such as S210E and S247E and test them in the same experimental system.

4. In the Figures 4A to C, the authors showed that Fgf4 KO ESCs can induce several differentiation markers but cannot upregulate Xist, suggesting Fgf4-mediated Yy1 pathway is required for Xist upregulation. However, they did not properly show whether pluripotency factors (Oct4/Nanog/Rex1) are downregulated upon RA treatment in these Fgf4 KO ESCs. Although they showed NANOG staining signal was weakened upon RA treatment (Fig. S4B), are Oct4 and Rex1 also downregulated with RA. As immunostaining is not quantitative, I recommend the authors to quantitate the expression levels of Oct4/Nanog/Rex1 by RT-qPCR.

5. In Figures 4D-F and S4C&D, all qPCR data to quantitate the gene expression lack the Day 0 data, making difficult to evaluate these data. The authors should add Day 0 values and use WT Day 0 as the relative expression "1.0" to show the relative expression level of the other samples at later time points. I think this modification could help to evaluate the relative change of each factor.

Minor points

6. Each cellular image should be presented as bigger size with scale bars. The current images are too small and difficult to see. They also have to explain clearly what the white arrowheads indicate in the figure legends.

7. It is strange to use a box plot to present luciferase assays (Fig. 3I). It should be presented by a bar chart like Fig. S3D. It is also necessary to show how many replicates were analyzed in this analysis.

8. Can YY1 S120A and S247A mutants be phosphorylated by ERK? If S247A YY1 mutant lacks phosphorylation by stimulated ERK, it could explain why S247A cannot activate Xist-promoter shown in Fig. 3I.

Referee #1:

In this manuscript, the authors explore the regulatory links between the initiation of X-chromosome inactivation (XCI) and cellular differentiation; while several molecular players are known to couple pluripotency with the timings of Xist expression (e.g., Nanog, Oct4), it remains unclear which global differentiation signalling pathways are involved. Here, the authors focus on FGF4 and downstream actors and show, through a series of loss-of-function and rescue experiments, that FGF4 is upstream of XCI initiation, through MEK/ERK signalling, which leads to phosphorylation of YY1, a known activator of Xist, and timely downregulation of pluripotency factors, which repress Xist expression. Despite the general roles of FGF4, the authors succeeded in pinpointing the specific pathways that are connected to the initiation of XCI. I consider this manuscript to be very relevant and significant, extending our knowledge of which pathways are involved in the initiation of XCI. The experimental approach is solid and elegant, including appropriate controls such as rescue experiments. I have no major concerns regarding this study, and I list below my minor concerns. Importantly, the manuscript should be overall improved for clarity (both in terms of language and methodological details, which are often lacking from the results description) and for consistency (there are sometimes discrepancies in numbers of figures/panels, but also in experimental details between results and methods sections).

Response: Thank you so much for your positive evaluation and insightful comments that have significantly improved the quality of our manuscript.

We have considered your comments carefully and revised the manuscript point by point. Additional experiments based on your comments have been performed; results and relevant discussion have been added in the revised manuscript. The revisions have been marked in red text within the revised manuscript, and the items that need to be addressed in detail are explained in the following point-by-point response to each comment.

With regard to the language and methodological details, thanks a lot for your detailed comments that make our manuscript clearer and easier to understand. We have refined our writing throughout the manuscript (**for details, see the point-by-point response below**). We apologize for the discrepancies and errors in the initial manuscript, we have checked each section to avoid these errors.

Minor concerns •

1. The very first results that link FGF signalling to random XCI are shown in supplementary - why not in the main figure? And their description in the text lacks methodological details - can the authors be more precise about what was done?

Response: Thanks so much for your insightful comments for enhancing our presentation. As you have pointed out, the results of FGFR blockage provided solid evidence that supports the essential role of FGF signaling in random XCI initiation, which will be impressive to readers. Thus, based on your comments, we have reorganized our results in the revised manuscript (**Fig 1A-C**). This reorganization allows readers to immediately appreciate the central premise of our study: FGF signaling serves as an essential molecular trigger for random XCI.

In addition, we have added necessary methodological details, experimental design, and processes in the Results section to make the results more informative and easier to understand. The added statements are presented as follows: “We exposed female ES cells to Infigratinib (BGJ398), a pan-FGFR inhibitor, and detected H3K27me3 state, the hallmark of XCI at day 5 of differentiation” (Lines 103-106).

2. The results regarding the functional window for FGF4 to initiate random XCI are very interesting. Have the authors confirmed that most cells still have two X chromosomes present after 3 days of differentiation without FGF4? If most cells have lost one X chromosome, adding FGF4 would not be expected to initiate XCI. Moreover, how healthy do the cells look after three days of differentiation without FGF4? Presumably, without XCI, XX cells would die during differentiation / not differentiate properly (Schulz et al. 2014) and/or lose an X chromosome (or XO cells are favoured). It could thus be that what the authors observed is not about a functional window for FGF4 to initiate XCI but an XX/XO issue. This alternative, if not addressed experimentally, should at least be mentioned in the text.

Response: Thanks for your detailed reminder that ensures our materials are reliable and suitable for the experimental design.

As you have mentioned, XO ES cells that have lost one X chromosome do not initiate XCI (Monkhorst et al, 2008) , thus we should first exclude the possibility that the failure of random XCI may be due to the loss of one X chromosome. Based on your comments, we have performed karyotype analysis. Both wild-type and *Fgf4*^{-/-} ES cells, at either undifferentiated or differentiating status, have two normal X chromosomes. In addition, no abnormalities have been observed in either X chromosomes or autosomes (see the figure below).

Figure. The karyotype analysis of WT, *Fgf4* KO-84# ES cells at the ground state, or at day 3 and 7 of differentiation. The boxed region highlighted by the red dotted line indicates X chromosomes.

We have added these results in the revised manuscript (Appendix Fig. S2D,E, Lines 129-131). In addition, as you mentioned above, in our initial manuscript, we found that adding exogenous FGF4 completely rescued *Xist* upregulation and coating, as well as restored H3K27me3 state to levels comparable to those in wild-type ES cells (Fig. 1F-J), so the

possible chromosomal loss may not have occurred.

To assess the health status of *Fgf4*^{-/-} ES cells, we have conducted flow cytometric analysis to measure the apoptotic rate (see the figure below). The results showed that the apoptotic rate of *Fgf4*^{-/-} ES cells, at either ground state or during differentiation, was comparable to those of wild-type ES cells. We have added these results in the revised manuscript (Appendix Fig. S2F,G, Lines 129-131).

Figure. Flow cytometric analysis of apoptosis by Annexin-V/PI double staining in WT, *Fgf4* KO-84# ES cells at the ground state or day 3 of differentiation.

These results were in line with previous observations that FGF4 is essential for regulating differentiation commitment, but not indispensable for the propagation and survival of ES cells (Kunath et al, 2007).

Taken together, our initial results and additional data show that FGF4's role in initiating random XCI appears not to be due to its effect on survival and differentiation. Moreover, karyotype analysis, and the rescue experiment of FGF4 supplementation in the initial manuscript, indicate that the significant loss of H3K27me3 and *Xist* domains is not due to the loss of X chromosome *per se*, but rather to the failure of random XCI. We have supplemented additional data, as well as relevant statements in the sections of Results (Lines 129-131, Appendix Fig. S2D,E) and Discussion (Lines. 267-272).

3. For the non-specialists, it is not clear at all why the experiments with heparin are relevant; the authors could first point out how heparin relates to FGF4.

Response: Thank you so much for your comments that pointed out our ambiguous statement. Previous studies have shown that the sulfate groups of heparin play important roles in promoting the heparin-dependent FGF4-FGFR interaction, and FGF4 needs to interact with heparin, serving as a cofactor, to exert its optimal biological activity (Aviezer et al, 1999;

Bellosta et al, 2001). Thus, in relevant studies, FGF4 supplementation experiments are always performed in the presence of heparin (Kang et al, 2017; Kang et al, 2013; Molotkov et al, 2017; Schrode et al, 2014; Yamanaka et al, 2010). We have added relevant statements in the revised manuscript to make it easier to understand (Lines 135-137).

4. Interesting that the authors observe the same effects in the rescuing experiments, whether they use FGF4 or FGF2. Is it known that FGF4 and FGF2 signal via the same receptor in mESC? Same downstream pathways? Adding explanations/clarifications to the text would be helpful.

Response: We apologize for our unclear statement. Several lines of evidence suggest that FGF4 and FGF2 may regulate ES cells through the same receptor: (1) FGF4/FGFR1 signaling is required for ES cell exit from pluripotency (Kang et al, 2017; Bellosta et al, 2001) ; (2) FGFR1 is the main receptor that responds to FGF2 in human ES cells (Dvorak et al, 2005) ; (3) FGF4 and FGF2 interact with FGFR1 during mesoderm induction, angiogenesis, or in somatic cells (Wang et al, 2004) .

In addition, in the previous studies, FGF2 has been frequently used as an alternative, but non-physiological FGF ligand (FGF2 is not expressed in the ICM and ground-state ES cells), to test phenotypes caused by *Fgf4* or *Fgfr* deficiency, such as primitive endoderm (PrE) specification, and obtained comparable results with exogenous FGF4 supplementation (Molotkov et al, 2017; Kang et al, 2013) .

We have added relevant statements in the revised manuscript to clarify the use of FGF2 (Lines 137-142).

5. How does FGF4 signalling (levels) relate to the differentiation status of mESC? Adding explanations/clarifications to the text would be helpful.

Response: Thank you for your detailed reminder to explain the association of FGF4 signaling and ES differentiation. Given FGF4 is a well-known autocrine signal driving ES cells exit from the pluripotent state, and considering random XCI only initiates upon the differentiation of ES cells, it is crucial, as you rightly pointed out, to emphasize the dynamics of FGF4 expression and the activity of the FGF4-FGFR signaling.

To this end, we reanalyzed the dynamic transcriptome studies of the early stages of ES differentiation. The reanalysis results of different data (Barros et al, 2019; Schulz et al, 2014) , along with the RT-qPCR detection, consistently showed that *Fgf4* is the primarily expressed ligand compared to other ligands. Both *Fgf4* and its major receptor *Fgfr1* are maintained at steady-state levels in the early phase of ES differentiation, by which rXCI is initiated. In addition, to test the activity of FGF4 signaling, we used a method of calculating ligand-receptor interaction scores (Kumar et al, 2018) and found that FGF4-FGFR1 interaction scores were highly active during the stage.

Figure. Dynamic expression levels of *Fgfs* and *Fgfrs* (upper panels), as well as the interaction score of *Fgf4* and its major receptors (lower panels) at different time points of differentiation using publicly available transcriptome dat.

Figure. The relative mRNA expression levels of *Fgf4*, *Fgf5*, and *Fgfr1* and *Fgfr2* genes in WT ES cells at days 0, 1, 3 and 5 of differentiation.

We have added and emphasized these data in the revised manuscript to highlight the association of FGF4 signaling with ES differentiation (Lines 115-120).

6. It would be helpful if the authors could describe in more detail the online tool they used (<http://egastrulation.sibcb.ac.cn>), particularly what the tool shows exactly. For instance, in the legend of Fig.1A, it says that "Each dot in the plot represents the cell sample at the specific positional address" - what does this mean exactly? What is the cell sample?

Response: Thanks so much for your detailed reminder to make our manuscript more informative.

The online tool (eGastrulation: <http://egastrulation.sibcb.ac.cn>) is a 3D gene expression database of gastrulating mouse embryos (~ E5.5 - E7.5) (Peng et al, 2016; Peng et al, 2019) . In brief, the embryos were serially cryo-sectioned and samples from different regions (ectoderm, mesoderm and endoderm) were collected. On each section, approximately 20 cells

were isolated by laser capture microdissection from the ectoderm, mesoderm and endoderm in each of the quadrants: anterior (A), posterior (P), and lateral (left/right, L/R), of each sample section (S1, S2, S3... etc, with section 1 being the most distal). The reference sections (R1, R2, R3... etc) were used as templates for 3D reconstruction of the embryo for data visualization (see the below schematic diagram).

Figure. Schematic illustration of construction of eGastrulation at E5.5 (A), E6.0 (B) and E6.5(C) in the online tool (eGastrulation). The right panel in each stage indicate the sample section. The captured regions are labelled in the section and corresponding abbreviations in each section are indicated below. (D)The presentation of the spatial pattern of gene expression in the corn plot (so named for its resemblance to a corn cob). Each dot in the plot represents the cell sample at the specific positional address, and the color indicates the level of gene expression computed from the transcript counts in the RNA-seq dataset. The position of each cell sample was represented by the grid node in a corn plot, which is defined by the coordinate position in the cross-sectional plane (A, P, R, L) and the distal-proximal axis (1–11).

Regarding the sentence you mentioned: "Each dot in the plot represents the cell sample at the specific positional address". We have recognized it may not fully or accurately convey the message, thus we have rewritten the sentence and added a schematic diagram to make it clearer and easier to understand. We have refined the figure and description as follows:

Figure. The presentation of the spatial pattern of gene expression of *Fgf4* and *Fgfr1* in gastrulating mouse embryos. The upper panel: the schematic diagram of E5.5, E6.0 and E6.5 embryos. The boxed region indicates the epiblast, in which random XCI occurs, and the schematic model (right) depicts the sample section and captured regions. The position of each sectional sample is labelled by the number on the distal-proximal axis. The captured regions are labelled in the section, and corresponding abbreviations in each section are indicated below. The lower panel: spatial expression pattern of *Fgf4* and *Fgfr1* in the epiblast at the indicated stage, which is presented as the corn plot (so named for its resemblance to a corn cob). Each dot in the plot represents the cell sample at the specific positional address indicated by the number next to the plot and the abbreviation at the top. The color indicates the level of gene expression computed from the transcript counts in the 3D gene expression database (eGastrulation: <http://egastrulation.sibcb.ac.cn>).

Together, we have added or refined relevant statements in the sections of Results, Figure legend (Lines 685-696, Figure 1D), as well as a subsection in the Materials and Methods (Lines 357-368), to make the results clearer and more informative.

7. The authors show that the downregulation of pluripotency factors is impaired upon inhibition or knockout of FGF4 and show expression data at D1, D3 and D5. Why not showing expression data at D0, which is the starting point? (the figure legend actually mentions D0, but the data is not shown in the figure). The timeline of FGF4 inhibition should also be indicated, i.e., when was the inhibitor added, especially in relation to start of differentiation?

Response: We apologize for our ambiguous presentation. The day when ES cells were transferred to the differentiation medium was designated as day 0. To enhance the

understanding of our results, we added the schematic diagram of the timeline below the X-axis to make our results easier to understand (see the figure below). In addition, we have refined the relevant statements in the Results section, as well as the Figure legend (Lines 776-780, Figure 4D).

Figure. Dynamic expression levels of *Nanog*, *Prdm14*, *Rex1* in WT, WT+FGFRi, *Fgf4* KO-84#, *Fgf4* KO-84#+FGF4 ES cells at days 0, 1, 3 and 5 of differentiation. The schematic diagram below the X-axis in D indicates the timeline of differentiation. The day when ES cells were transferred to the differentiation medium was designated as day 0.

8. In general, the manuscript would benefit from including more methodological explanations along with the description of the results; I have already provided some examples. Another one would be the section on the experiments with mice, the authors could make understanding easier by including more details.

Response: Thanks so much for your detailed reminder to make our manuscript more informative and easier to understand. We have added essential details and rewritten the relevant sentence in the *in vivo* experiments and reorganized the schematic diagram of the experimental design and workflow (Lines 170-173; 730-733; Figure 2H). In addition, based on your comments, we have checked the Results section thoroughly and added necessary details in the Results section (Lines 103-105; 115-120; 123-124; 157-158; 170-173; 191-192; 197-198).

Figure. Schematic diagram of the experimental workflow. Pregnant mice received daily intraperitoneal injections of the MEK/ERK inhibitor, FGFR inhibitor or in combination of ERK-activating agent PMA, during the window of random XCI. The female epiblast was collected and tested for *Xist* expression and iXCI status.

9. The letters on top of the barplots to represent statistical significance is not the most usual representation. It can be maintained if according to the journal's policies, otherwise a small table below the barplot could be used, or the more usual stars and lines specifying which comparisons were made.

Response: Thanks so much for your detailed reminder to make the comparison clearer and easier to understand. As you have pointed out, stars and lines on the top of bar charts are a better choice when showing comparisons of the two specified groups. We have checked the results throughout the manuscript and refined the presentation of relevant results (**Figure 1K; 2B,C,E,F;3D,F,I-K;4B,C**).

10. Corrections: One of the Ser→Ala substitutions is referred to being at either position 118 (text) or position 120 (text and figure).

Response: We apologize for the writing error due to our carelessness. We have corrected the information in the revised manuscript (**Figure 3H, I and the corresponding figure legend, Line 756-765**).

11. Corrections: In the Results, the mutations of YY1 are referred to as Ser→Ala substitutions, but in the Methods they are referred to Ser→Asp.

Response: Thanks so much for your detailed reminder, we have corrected the writing error (**Lines 208; 451**). In addition, based on the comments of reviewer 2, we have supplemented additional data by substituting serine 120 and 247 with aspartic acid (Asp, D), which is frequently used to mimic the phosphorylated state, in the revised manuscript (**Line 211-221; 451**).

12. Figures 3H and 3I could be improved. In Fig. 3H it is confusing that the YY1 protein structure is shown below the Xist locus, and with the same length. In Fig. 3I the authors depict YY1 motifs with different shapes for wildtype and mutant YY1 forms, but the motifs are always the same, so I believe they should be represented by the same shape.

Response: We apologize for our ambiguous presentation that makes our results difficult to understand. We have reorganized our results as follows:

(1) Remove the map of the *Xist* 5' region containing YY1 motifs to make the schematic diagram depicting the strategy of Ser→Ala and Ser→Asp clearer and more concise (**Figure 3I**).

Figure. Schematic diagram of DNA mutations via the substitution of serine (Ser, S) 120 and 247 with alanine (Ala, A, non-phosphorylation-mimic mutation) or aspartic acid (Asp, D, phosphorylation-mimic mutation) with the coding region of *Yy1* gene. For the Ser→Ala substitution at the positions 120 or 247, the triplet codons TCG and TCA were respectively replaced by CGG and CGA; For the Ser→Asp substitution at the position 120 or 247, the triplet codon TCG and TCA were replaced by CTG and TCG, respectively.

(2) Reorganize the result of luciferase assay using bar charts labeled with + and – (Figure 3I).

Figure. Relative luciferase activity of the reporters containing Xist promoter, which were co-transfected with an empty vector (pCAGGS), or overexpression vector of WT or mutant *Yy1* in WT ES cells. Each dot indicates the result of each reaction, and assays were replicated independently at least three times.

13. Corrections: Some supplementary figures are not well indicated in the main text (e.g., S1N should be S1O; S1L should be S1N; S1J-K should be S1K-L).

Response: Thanks so much for your detailed reminder, we apologize for our carelessness. We have corrected these errors and checked the figure citation throughout the manuscript.

14. Corrections: Some figure legends have mistakes regarding the letters of the panels (e.g., Fig. 1).

Response: We apologize for our carelessness. We have corrected these errors and checked the figure citation throughout the manuscript.

15. References:

i. "Initiation of random XCI strictly relies on the upregulation of long noncoding RNA Xist upon differentiation from the pluripotent state (Makhlouf et al, 2014)" This was first shown in Penny et al. 1996, not Makhlouf et al. 2014.

Response: The reference has been replaced with “Penny GD, Kay GF, Sheardown SA, Rastan S, Brockdorff N. Requirement for Xist in X chromosome inactivation. Nature. 1996 Jan 11;379(6561):131-7. doi: 10.1038/379131a0. PMID: 8538762”.

ii. I was surprised that the authors never mentioned the work from Schulz et al. 2014, which revealed feedback regulation between MAPK signaling and initiation of X-inactivation

Response: We apologize for missing this important content. As you pointed out, the study by Schulz et al. provided a mechanistic link between the X chromosome dose and developmental progression by elucidating that two active X chromosomes stabilize the naïve pluripotent state by inhibiting the ERK pathway, thus delaying ES differentiation (Schulz et al, 2014) . By contrast, our results identify that once exiting the pluripotent state, activated ERK reciprocally triggers random XCI. Thus, Schulz’s results and our findings support the reciprocal antagonism of ERK and X chromosome dose. We have cited Schulz’s work in the Discussion section and elaborated on this point (Lines 290-295).

iii. "A variety of positive and negative factors ensuring Xist upregulation have been described: transcription factors YY1 and GATA can directly drive robust Xist transcriptional activation (Jonkers et al, 2009); RNF12, an E3-ubiquitin ligase can target and degrade REX1, a critical repressor of Xist, acts as a dose-dependent activator of Xist upregulation (Navarro et al, 2008; Tian et al, 2010)." The references seem to be mixed up. Jonkers et al. has not shown the regulation of Xist by YY1 and GATA; these findings were made by Makhlouf et al. 2014 and by Lustig et al. 2023. The role of RNF12 was instead shown by Jonkers et al. 2009.

Response: We apologize for the issues caused by an operational error in the reference management software. We have corrected the references and thoroughly reviewed the manuscript to address related concerns.

16. The second paragraph in the discussion would have been helpful in the introduction already; it could be moved.

Response: Thanks so much for your comment, which helped us to optimize the logical relationship between the Introduction and Discussion sections. We have moved this content to the Introduction section and rewritten these sentences to maintain logical consistency with the existing content in the Introduction (Lines 57-59; 71-74).

17. Language/writing could be improved throughout, especially for clarity.

Response: We greatly appreciate your reminder. We have reviewed the language and writing throughout the manuscript to enhance the clarity and comprehensibility of our statements.

18. Not clear what "holistically" in the title means. The syntagm "loss of pluripotency factors" is not well linked with the rest.

Response: Thank you for your detailed comments that have made our expressions more accurate and avoided ambiguity. We have rewritten relevant sentence in the revised manuscript as follows:

In the Abstract section, “we show how FGF4 holistically regulates XCI and differentiation” has been rewritten as “we show how FGF4 comprehensively orchestrates XCI and differentiation”,

In the Discussion section, “via the holistic governance of the opposing pathways” has been rewritten as “via the comprehensive regulation of the opposing pathways”

In addition, as you have suggested, we have removed the sentence “even when *Xist* is derepressed upon the loss of pluripotency factors” since this sentence is not highly relevant to the following content, nor is it necessary here.

Referee #2:

Comments to the authors

In this manuscript, Ma et al. investigate molecular mechanisms of *Xist* upregulation, which triggers initiation of random XCI in female mammals. Based on analyses with mouse female ESC differentiation system, they showed that (1) FGF4 is critical for *Xist* upregulation upon differentiation and (2) phosphorylation of YY1 by FGF4-ERK pathway is important for direct upregulation of *Xist*. The authors propose that these two pathways contribute to upregulate *Xist* in parallel. I think their data well supports (1) although it is already accepted that FGF4 is critical for the induction of differentiation in mouse ESCs. Their data supporting (2) is, however, not strong and need to be more strengthened by additional experiments listed below. My concern is FGF4 signal may contribute simply to ESC differentiation and indirectly upregulate *Xist* expression, but not much contribute to YY1 phosphorylation-mediated direct upregulation of *Xist*.

Response: Thanks so much for your insightful and constructive comments that are greatly helpful for improving the quality of our manuscript. Based on these comments, we have (1) supplemented additional data to strengthen the evidence supporting our findings (for details, see the response to the third major point below); (2). refined our statements to enhance the precision and clarity of our conclusions, and in particular, discussed the data about the phosphorylation-stimulating effect of FGF4-ERK on YY1 cautiously and avoided overstatement .

Please find below our point-by-point responses.

Major points:

1. First of all, there are many typos in the figure legends. For example, all statistics should be indicated with "*", not with alphabets such as "a" or "b". In addition, description in the figure legends is not enough to understand each figure. The authors should provide more information in the legend to understand each figure such as experimental conditions and explanation for abbreviations. These figure legends made the presented data not easy to understand. The authors should also provide more information in the methods. For example, it is not clear which ESC line they used in this study. Before submission of manuscript, these points have to be carefully checked and corrected.

Response: Thank you for your detailed reminder. As you have pointed out, using stars and

lines on the top of bar charts is a better choice when showing comparisons of the two specified groups. We have revised all such issues (Figure 1K; 2B,C,E,F;3D,F,I-K;4B,C).

In addition, we have checked the results throughout the manuscript and refined the presentation of relevant figures and the figure legend to make them clearer and easier to understand. For example, the figures 3H-I in the initial manuscript have been reorganized to clearly present the experimental strategy and experimental conditions for the results of the luciferase assay (Figure 3H-K); A schematic diagram has been added to clarify the spatial pattern of gene expression in the corn plot (Figure 3H-K); Some results have been labeled with the ES cell line used, as well as a schematic diagram of the timeline of ES cell differentiation (Figure 3I), and so on.

With regard to the figure legend and the statement of methodology, we apologize for our unclear or even ambiguous statements. During the revision, we checked these issues throughout the manuscript, and supplemented or revised the captions and method descriptions in all figure legends, and the Materials and Methods sections (Line 337-342; 357-368; 384-391).

2. The authors proposed that phosphorylated form of YY1 activates *Xist* transcription. Is YY1 phosphorylated at which time points during ESC differentiation? A time-course experiment showing YY1 phosphorylation during differentiation should be presented like Fig. S2B&C. As YY1 has been shown to be important for sustained expression of *Xist* in MEFs (Makhlouf 2014), I think it is important to examine whether YY1 is phosphorylated only transiently at the beginning of differentiation or during the whole differentiation time-windows.

Response: Thank you for your insightful comments, which enhance the support for our key findings. As you suggested, the result of dynamics of YY1 phosphorylation will essentially improve our understanding of YY1's role in upregulating *Xist* and initiating random XCI. The results of Phos-tag-based Western blot showed that YY1 phosphorylation levels, above the basal level, were transiently increased at the window of differentiation initiation, approximately in line with the ERK phosphorylation activation during the initial differentiation (see the figure below).

Figure. Phos-tag analysis of pYY1 and Western blot analysis of β -TUBULIN in WT ES cells at day0-0h, day0-2h, day 1, day 3 and day 5 of differentiation.

We have added these results in the revised manuscript (Figure 3A, Lines 194-198), and elaborated on the involvement of phosphorylated YY1 in *Xist* upregulation and XCI initiation upon the differentiation in the Discussion section (Lines 280-286).

3. The authors examined the importance of YY1 phosphorylation by mutating putative

phosphorylation sites of YY1 and showed that S247A mutation lacks the activation of Xist-promoter-luciferase activity (Fig 3I). My concern is that even WT YY1 was able to activate luciferase activity about only 2-fold, which is quite weak. Can the authors examine the effect of YY1 mutant overexpression on endogenous Xist expression? I think this experiment is important to strengthen their hypothesis that phosphorylation of YY1 facilitates Xist expression upon differentiation. Ideally, endogenous Yy1 should be knocked out, and then exogenous Yy1 WT, S120A, and S247A will be expressed to see how much Xist can be transcribed. I also would like the authors to create the phosphorylation mimic form of YY1 mutants, such as S210E and S247E and test them in the same experimental system.

Response: Thank you for your constructive comments to help improve our experimental design. The extent of increase in luciferase activity is likely positively correlated with the efficiency of YY1 overexpression, which can be explained by our additional experiments (see below).

We fully agree with your view and recognize the need to supplement relevant experiments to detect the effect of exogenous *Yy1* mutant overexpression on endogenous *Xist* expression. Thus, we overexpressed wild-type (WT) or mutant *Yy1*, and detected the endogenous *Xist* expression. Our data demonstrated that, unlike the significant elevation of *Xist* expression caused by WT *Yy1* overexpression, either S247A or S120A mutation (Ser→Ala substitution at position 247 or 120 to mimic the non-phosphorylated state) markedly abolished this inductive effect of *Yy1*. Conversely, overexpression of mutant *Yy1* with the substitution of serine 120 and 247 with aspartic acid (Asp, D), which were frequently used to mimic the phosphorylated state (Kassardjian et al, 2012; Li et al, 2004; Trautwein et al, 1993), resulted in comparable *Xist* expression levels with that of WT *Yy1*.

Combined with the luciferase assay results, these findings underscored the important role of YY1 phosphorylation in driving *Xist* expression. We have supplemented the additional data in the revised manuscript (see below. Figure 3H-K; Lines 211-220 in the revised manuscript).

Figure. (A) Schematic diagram of DNA mutations via the substitution of serine (Ser, S) 120 and 247 with alanine (Ala, A, non-phosphorylation -mimic mutation) or aspartic acid (Asp, D, phosphorylation-mimic mutation) with the coding region of *Yy1* gene. For the Ser→Ala substitution at the positions 120 or 247, the triplet codons TCG and TCA were respectively replaced by CGG and CGA; For the Ser→Asp substitution at the position 120 or 247, the triplet codon TCG and TCA were replaced by CTG and TCG, respectively. (B) Relative luciferase activity of the reporters containing *Xist* promoter, which were co-transfected with an empty vector (pCAGGS), or overexpression vector of WT or mutant *Yy1* in WT ES cells. Each dot indicates the result of each reaction, and assays were replicated independently at least three times. (C and D) Relative expression levels of endogenous *Yy1* (J) and *Xist* (K) in WT ES cells transfected with an empty vector (pCAGGS), or overexpression vector of WT or mutant *Yy1* in WT ES cells.

Moreover, we further adopted your suggestion by performing *Yy1* knockout (KO) and subsequently detected the effects of exogenous overexpression of WT or mutant *Yy1* on *Xist* expression. However, during the screening for stable KO cell lines, we observed that *Yy1* deficiency resulted in severely impaired cell viability and proliferation, making it difficult to obtain the desired *Yy1* KO cell lines. This finding might align with the previous notion that YY1 is required for the stemness potential of ES cells (Martinez-Ruiz et al, 2021), probably due to the role of YY1 in mediating transcriptional activation of *Oct4* and *Sox2* (Gao et al, 2013) or

maintaining posttranscriptional stability of SOX2 and OCT4 proteins (Wallingford et al, 2017) . Based on these considerations, and given that existing WT cell lines have already provided direct evidence, we have not provided additional evidence from KO cell lines. We sincerely hope for your understanding regarding this decision.

With regard to the experiments of mimicking phosphorylated YY1, it is a constructive comment to improve our experimental design and further support our findings. Thus, during this revision, we have constructed a mutant *Yy1* in which both serine 120 and 247 were substituted with aspartic acid (Asp, D), which were frequently used to mimic the phosphorylated state (Kassardjian et al, 2012; Trautwein et al, 1993; Li et al, 2004) . Luciferase assays showed that the substitution of either Ser 120 or 247 with Asp showed comparable transcriptional activity with WT YY1. These results were also confirmed in 293T cells. We have supplemented additional data in the revised manuscript (see below. Figure 4I, Appendix Fig. S5D; Lines 211-220 in the revised manuscript).

Figure. Relative luciferase activity of the reporters containing *Xist* promoter, which were co-transfected with an empty vector (pCAGGS), or overexpression vector of WT or mutant *Yy1* in WT ES cells. Each dot indicates the result of each reaction, and assays were replicated independently at least three times (left). Relative luciferase activity of the reporters containing the *Xist* promoter, which were co-transfected with an empty vector (pCAGGS), or the overexpression vector of WT or mutant *Yy1* in 293T cells. Each dot indicates the result of each reaction and assays were replicated independently at least three times (right).

Together, thanks to your constructive comments that improve our experimental design, we have supplemented the results of endogenous *Xist* expression induced by exogenous WT or

mutant *Yy1*, as well as luciferase assays by using phosphorylation and non-phosphorylation -mimic mutation, to make our data more solid. We have also refined the statements in the Discussion section, to enhance the precision and clarity of our conclusions and, in particular, avoid overstatement (Line 280-289).

4. In the Figures 4A to C, the authors showed that *Fgf4* KO ESCs can induce several differentiation markers but cannot upregulate *Xist*, suggesting *Fgf4*-mediated *Yy1* pathway is required for *Xist* upregulation. However, they did not properly show whether pluripotency factors (*Oct4*/*Nanog*/*Rex1*) are downregulated upon RA treatment in these *Fgf4* KO ESCs. Although they showed NANOG staining signal was weakened upon RA treatment (Fig. S4B), are *Oct4* and *Rex1* also downregulated with RA. As immunostaining is not quantitative, I recommend the authors to quantitate the expression levels of *Oct4*/*Nanog*/*Rex1* by RT-qPCR.

Response: Thanks for your detailed comments that make our data more solid. As you pointed out, the changes of core pluripotency factors upon RA treatment in the *Fgf4* KO ES cells are important to support our notion that enforced induction of differentiation *per se* is not enough to rescue *Fgf4*^{-/-} cells from XCI failure. Thus, based on your comments, we have detected the expression dynamics of the aforementioned pluripotency factors. The additional data showed that RA facilitated the decline of *Nanog*, *Prdm14* and *Rex1*, but not *Sox2* and *Oct4* (see the figure below). These findings are in line with the results of delayed decline of core pluripotency factors in *Fgf4* KO or FGFR-inhibited WT cells. We have added these results in the revised manuscript as evidence of RA's differentiation-inducing effect (Lines 228-229) and discussed their relevance to the contribution of the timely decline of pluripotency factors to random XCI initiation (Appendix Fig. S6B, Lines 306-309).

Figure. Dynamic expression levels of *Nanog*, *Prdm14*, *Rex1*, *Sox2*, *Oct4* in *Fgf4* KO-84#, *Fgf4* KO-84#+RA ES cells at day 0, 1, 3 and 5 of differentiation.

5. In Figures 4D-F and S4C&D, all qPCR data to quantitate the gene expression lack the Day 0 data, making difficult to evaluate these data. The authors should add Day 0 values and use WT Day 0 as the relative expression "1.0" to show the relative expression level of the other samples at later time points. I think this modification could help to evaluate the relative change of each factor.

Response: Thanks so much for your detailed reminder. We apologized for our ambiguous presentation. The day when ES cells were transferred to the differentiation medium was designated as day 0. We have added the expression levels of Day 0 and, as you suggested, normalized the value of WT Day 0 to 1. In addition, to enhance the understanding of our results, we've added a schematic diagram of the timeline below the X-axis (see the figure below).to make our results easier to understand Moreover, we have refined the relevant statements in the Results section, as well as the Figure legend (**Figure 4D-F, Appendix Fig. S6B,D, Lines 240-243; 778-780**).

Figure. Dynamic expression levels of *Nanog*, *Prdm14*, *Rex1*, *Sox2* and *Oct4* in WT, WT+FGFRi, *Fgf4* KO-84#, *Fgf4* KO-84#+FGF4 ES cells at days 0, 1, 3 and 5 of differentiation. The schematic diagram below the X-axis in D indicates the timeline of differentiation. The day when ES cells were transferred to the differentiation medium was designated as day 0.

Minor points

6. Each cellular image should be presented as bigger size with scale bars. The current images are too small and difficult to see. They also have to explain clearly what the white arrowheads indicate in the figure legends.

Response: We apologize for our improper presentation. We have enlarged relevant figures and added scale bars in each figure. In addition, we have labelled arrowheads in all relevant figures, and added the explanation in the figure legend to clarify the white arrowheads.

Moreover, based on your reminder, we have refined and enriched the content of the figure legend throughout the manuscript, making them clearer and easier to understand.

7. It is strange to use a box plot to present luciferase assays (Fig. 3I). It should be presented by a bar chart like Fig. S3D. It is also necessary to show how many replicates were analyzed in this analysis.

Response: Thanks for your reminder. We have reorganized the results of luciferase assays as a bar chart with scatter points, and rewritten the figure legend to clarify the technical and independent biological replicates (The legend of Figure 3I). In addition, We have supplemented the figure legends with information on replicate numbers throughout the manuscript.

8. Can YY1 S120A and S247A mutants be phosphorylated by ERK? If S247A YY1 mutant lacks phosphorylation by stimulated ERK, it could explain why S247A cannot activate Xist-promoter shown in Fig. 3I.

Response: As you have pointed out, the phosphorylated state of serine 120 and 247 or their mutants will be very helpful to support our findings. Unfortunately, there are currently no commercially available specific antibodies targeting this phosphorylation site, so we are unable to show its phosphorylation status under various experimental conditions. We sincerely appreciate your understanding.

However, our data from both luciferase assays and the detection of endogenous Xist expression after overexpression using phosphorylation and non-phosphorylation-mimic mutations have shown the functional link between the phosphorylation of these sites and the transcriptional activity of YY1, as well as *Xist* expression. In addition, it is noteworthy that the function of these phosphorylated sites in affecting YY1's transcriptional activity has also been reported previously in other cell types(He et al, 2010; Kassardjian et al, 2012; Martinez-Moreno et al, 2017), suggesting that the effect may be relatively common across different cell types.

We have addressed this issue in the Discussion section (Lines 280-289).

References:

Aviezer D, Safran M, Yaron A (1999) Heparin differentially regulates the interaction of fibroblast growth factor-4 with FGF receptors 1 and 2. *BIOCHEM BIOPH RES CO* **263**(3): 621-626

Barros DAES, Jonkers I, Syx L, Dunkel I, Chaumeil J, Picard C, Foret B, Chen CJ, Lis JT, Heard E, Schulz EG, Marsico A (2019) Kinetics of Xist-induced gene silencing can be predicted from combinations of epigenetic and genomic features. *GENOME RES* **29**(7): 1087-1099

Bellosta P, Iwahori A, Plotnikov AN, Eliseenkova AV, Basilico C, Mohammadi M (2001) Identification of receptor and heparin binding sites in fibroblast growth factor 4 by structure-based mutagenesis. *MOL CELL BIOL* **21**(17): 5946-5957

Dvorak P, Dvorakova D, Koskova S, Vodinska M, Najvirtova M, Krekac D, Hampl A (2005) Expression and potential role of fibroblast growth factor 2 and its receptors in human embryonic stem cells. *STEM CELLS* **23**(8): 1200-1211

Gao F, Wei Z, An W, Wang K, Lu W (2013) The interactomes of POU5F1 and SOX2

enhancers in human embryonic stem cells. *SCI REP-UK* **3**: 1588

He Y, Kim JY, Dupree J, Tewari A, Melendez-Vasquez C, Svaren J, Casaccia P (2010) Yy1 as a molecular link between neuregulin and transcriptional modulation of peripheral myelination. *NAT NEUROSCI* **13**(12): 1472-1480

Kang M, Garg V, Hadjantonakis AK (2017) Lineage Establishment and Progression within the Inner Cell Mass of the Mouse Blastocyst Requires FGFR1 and FGFR2. *DEV CELL* **41**(5): 496-510

Kang M, Piliszek A, Artus J, Hadjantonakis AK (2013) FGF4 is required for lineage restriction and salt-and-pepper distribution of primitive endoderm factors but not their initial expression in the mouse. *DEVELOPMENT* **140**(2): 267-279

Kassardjian A, Rizkallah R, Riman S, Renfro SH, Alexander KE, Hurt MM (2012) The transcription factor YY1 is a novel substrate for Aurora B kinase at G2/M transition of the cell cycle. *PLOS ONE* **7**(11): e50645

Kumar MP, Du J, Lagoudas G, Jiao Y, Sawyer A, Drummond DC, Lauffenburger DA, Raue A (2018) Analysis of Single-Cell RNA-Seq Identifies Cell-Cell Communication Associated with Tumor Characteristics. *CELL REP* **25**(6): 1458-1468

Kunath T, Saba-El-Leil MK, Almousaillekh M, Wray J, Meloche S, Smith A (2007) FGF stimulation of the Erk1/2 signalling cascade triggers transition of pluripotent embryonic stem cells from self-renewal to lineage commitment. *DEVELOPMENT* **134**(16): 2895-2902

Li CL, Lu CY, Ke PY, Chang ZF (2004) Perturbation of ATP-induced tetramerization of human cytosolic thymidine kinase by substitution of serine-13 with aspartic acid at the mitotic phosphorylation site. *BIOCHEM BIOPH RES CO* **313**(3): 587-593

Martinez-Moreno M, O'Shea TM, Zepecki JP, Olaru A, Ness JK, Langer R, Tapinos N (2017) Regulation of Peripheral Myelination through Transcriptional Buffering of Egr2 by an Antisense Long Non-coding RNA. *CELL REP* **20**(8): 1950-1963

Martinez-Ruiz GU, Morales-Sanchez A, Pacheco-Hernandez AF (2021) Roles Played by YY1 in Embryonic, Adult and Cancer Stem Cells. *STEM CELL REV REP* **17**(5): 1590-1606

Molotkov A, Mazot P, Brewer JR, Cinalli RM, Soriano P (2017) Distinct Requirements for FGFR1 and FGFR2 in Primitive Endoderm Development and Exit from Pluripotency. *DEV CELL* **41**(5): 511-526

Monkhorst K, Jonkers I, Rentmeester E, Grosveld F, Gribnau J (2008) X inactivation counting and choice is a stochastic process: evidence for involvement of an X-linked activator. *CELL* **132**(3): 410-421

Peng G, Suo S, Chen J, Chen W, Liu C, Yu F, Wang R, Chen S, Sun N, Cui G, Song L, Tam PP, Han JD, Jing N (2016) Spatial Transcriptome for the Molecular Annotation of Lineage Fates and Cell Identity in Mid-gastrula Mouse Embryo. *DEV CELL* **36**(6): 681-697

Peng G, Suo S, Cui G, Yu F, Wang R, Chen J, Chen S, Liu Z, Chen G, Qian Y, Tam P, Han JJ, Jing N (2019) Molecular architecture of lineage allocation and tissue organization in early mouse embryo. *NATURE* **572**(7770): 528-532

Schrode N, Saiz N, Di Talia S, Hadjantonakis A (2014) GATA6 levels modulate primitive endoderm cell fate choice and timing in the mouse blastocyst. *DEV CELL* **29**(4): 454-467

Schulz EG, Meisig J, Nakamura T, Okamoto I, Sieber A, Picard C, Borensztein M, Saitou M, Bluthgen N, Heard E (2014) The two active X chromosomes in female ESCs block exit from the pluripotent state by modulating the ESC signaling network. *CELL STEM CELL* **14**(2): 203-216

Trautwein C, Caelles C, van der Geer P, Hunter T, Karin M, Chojkier M (1993) Transactivation by NF-IL6/LAP is enhanced by phosphorylation of its activation domain. *NATURE* **364**(6437): 544-547

Wallingford MC, Hiller J, Zhang K, Mager J (2017) YY1 Is Required for Posttranscriptional Stability of SOX2 and OCT4 Proteins. *CELL REPROGRAM* **19**(4): 263-269

Wang S, Ai X, Freeman SD, Pownall ME, Lu Q, Kessler DS, Emerson CJ (2004) QSulf1, a heparan sulfate 6-O-endosulfatase, inhibits fibroblast growth factor signaling in mesoderm induction and angiogenesis. *PNAS* **101**(14): 4833-4838

Yamanaka Y, Lanner F, Rossant J (2010) FGF signal-dependent segregation of primitive endoderm and epiblast in the mouse blastocyst. *DEVELOPMENT* **137**(5): 715-724

Dear Dr Tian,

Thank you for submitting your revised manuscript (EMBOJ-2024-118190R) to The EMBO Journal. Please accept my sincere apologies for getting back to you with this unusual delay due to protracted referee input as well as detailed discussion in the editorial team. Your amended study was sent back to the referees for their scientific reassessment, and we have received reports from both, which I enclose below. As you will see, the experts state that the work has been substantially enhanced by the revisions and they are now in favour of publication, pending appropriate amendments.

We can thus invite you to address the remaining concerns raised by the referees regarding conclusiveness of claims related to YY1 phosphorylation by complementary experimental data, or adjusting the discussion of the findings and introducing textual caveats in the manuscript where appropriate.

Also, we now need you to take care of a number of issues related to formatting and data presentation as detailed below, which should be addressed at re-submission.

Please contact me at any time if you have additional questions related to below points.

Thank you for giving us the chance to consider your manuscript for The EMBO Journal. I look forward to your final revision.

Again, please contact me at any time if you need any help or have further questions.

Kind regards,

Daniel Klimmeck

>> Please add up to five keywords to your study.

>> Author Contributions: Remove the author contributions information from the manuscript text. Note that CRediT has replaced the traditional author contributions section as of now because it offers a systematic machine-readable author contributions format that allows for more effective research assessment. and use the free text boxes beneath each contributing author's name to add specific details on the author's contribution.

More information is available in our guide to authors.

>> Provide a 'Disclosure and Competing Interests Statement' after the Acknowledgements.

>> Section order should be corrected as follows: title page with complete author information, abstract, keywords, introduction, results, discussion, methods, data availability section, acknowledgements, disclosure and competing interests statement, references, main figure legends, tables, expanded figure legends.

>> References: adjust reference format to EMBO Journal format, 10 authors et al, and place References after the Discussion, before figure legends.

>> Add a Reagents and Tools table to the Methods section, as a separate file using the existing template in the Guide For Authors, listing key reagents, experimental models, software and relevant equipment.

>> Please remove the short-sentence bullet points from the manuscript text and provide as separate text file.

>> Funding: Please provide consistent funding information: enter the following funding information into our online system: '31930103'. Enter '2017YFD0501901' in the Acknowledgements.

>> Appendix file with ToC: the reagents and tools table should be uploaded as a separate file and using our template. The correct nomenclature for figures and tables is "Appendix Figure S1" etc. and "Appendix Table S1", this will need to be corrected in the appendix figure legends and table of contents. Page numbers should be added to the table of contents and the red font be removed in the final version.

>> Data availability section: please add a sentence "No datasets amenable to large-scale data repository deposition were generated in this study".

>> Please cite your earlier 2020 study (PMID: 33061820) in the Methods section.

>> Author Checklist:

>>>> Data availability section: remove indication 'primary datasets deposited'.

>>>> Data availability section: remove indication 'data citation in the reference list'.

>> Consider additional changes and comments from our production team as indicated below:

- Figure legends:

1. Please note that the exact p values are not provided in the legends of figures 1C, 2G, I, J-L; 3D, F, G, I, J, K; 4B, S3 B-F; S5 D; S6A.
2. Please indicate what */ **/ ***/ **** represents; if this represents p value(s), please specify the exact p value in the legend(s) of figure(s) 1L, 4C
3. Please note that in figure S5 D there is a mismatch between the annotated p values in the figure legend and the annotated p values in the figure file that should be corrected.
4. Please note that information related to n is missing in the legend of figure S6 A
5. Please note that the scale bar needs to be defined for figure S6 C
6. Please note that the white arrows are not defined in the legend of figure 1I, 2A, J, L; 4A, G; S4 D. This needs to be rectified.

Referee #1:

The authors have done a good job in addressing the reviewers' requests, mine included. Yet, their main claim on how the FGF4 pathway regulates Xist expression (and XCI) via both YY1 phosphorylation and pluripotency factors still lacks substantial back-up, especially regarding the YY1 phosphorylation.

1/ The new data presented with the YY1 mutants is still not sufficiently convincing, since the effects on Xist expression levels are relatively low. For instance, it is not clear whether Xist levels upon overexpression of the no-phosphorylation mutants are significantly different from the phosphorylation-mimic mutants (it does not seem to be the case). It might be that the effects would only be observed in a background without wildtype YY1, as suggested by the other reviewer. The authors report that YY1-knockout ES cells are not viable, so probably a YY1-degron would be necessary to address this point fully.

2/ The data presented to justify their claim is mostly correlative and lacks further controls. Examples: the increase in YY1-phosphorylation coinciding with the time window of XCI-induction (D1 of differentiation); is this effect specific to XX cells? The lower binding of YY1 to the Xist promoter in Fgf4-KO - is this specific to the Xist promoter, or is there lower binding in general for YY1 sites that are more differentiation-specific? (and would also be affected given the altered differentiation status of Fgf4-KO)

3/ The experiments with retinoic acid (RA) seem to suggest that the effects of Fgf4-KO on Xist expression are not solely due to impaired differentiation (thus highlighting other mechanisms involved, indirectly supporting the YY1-phosphorylation hypothesis). However, the authors do not show whether in the RA treatment for Fgf4-KO cells the levels of pluripotency factors are

sufficiently low, similar to wildtype or to when FGF4 is supplemented (and rescues) Xist expression. In other words, could it be that the RA treatment does not rescue Xist expression levels because the pluripotency factors are still too high?

In the absence of further evidence, the authors could still present their data as it is but would have to drop the claim about YY1 phosphorylation and change the manuscript accordingly.

The manuscript still needs substantial editing for clarity and consistency; maybe this can be provided by the journal.

Referee #2:

I thank the authors to have responded to most of my concerns. I think the revised manuscript has much more clear messages to readers. I appreciate that they made efforts to strengthen the data supporting phosphorylated Yy1 could up-regulate Xist (Figure 3). I still have several minor comments listed below about the newly added data in the revised manuscript. As the manuscript still contains lack of experimental conditions in several figures, e.g. Figures 3 and 4, the authors should carefully check the figure legend and Methods section.

1. Lines 131-135

"More importantly, we found that the addition of exogenous FGF4 to the differentiation medium, which compensates for the endogenous loss of FGF4, completely rescued Xist upregulation and coating, as well as the H3K27me3 state, to levels comparable to those in wild-type ES cells (Fig 1I,J; Appendix Fig. S3A)." Does Appendix Fig. S3A correspond to this result? It looks just showing the lack of ERK1/2 phosphorylation in FGF4 KO ESC lines. If so, this data should be placed in Fig S2, between the panels C and D.

2. Fig. 3H-K

I thank the authors for adding these new data using phospho-mimic Yy1 and endogenous Xist promoter. Although the results support the significant role of Yy1 phosphorylation to up-regulate Xist, the degree of up-regulation (~2-fold) is weaker than endogenous Xist up-regulation observed in differentiating ESCs (usually > 10-fold upon differentiation). This would suggest the necessity of other factor(s) other than phosphorylated Yy1 to fully up-regulate Xist. I recommend the authors to add a few sentences to describe this point in the text.

3. Appendix Fig. S1B, S3A,E and S4B,C

I thank the authors to add Yy1 phosphorylation time-course data (Fig 3A). This raises a new question on how Yy1 is phosphorylated only upon differentiation. FGF4 is already expressed in undifferentiated WT ESCs (Fig S1B 0 hour, Fig S1D D0) but does not induce Xist up-regulation, which is consistent with the fact that ERK1/2 are not yet phosphorylated in undifferentiated state (Fig S4B,C). ERK1/2 are, however, rapidly phosphorylated in 0.5 hour upon differentiation (Fig S4B,C), which is followed by phosphorylation of Yy1 at Day 1 of differentiation (Fig 3A). These observations indicate the importance of differentiation itself to induce a rapid ERK1/2 phosphorylation. How this rapid phosphorylation of ERK1/2 upon differentiation could happen? As the exogenous addition of FGF4 increases Xist expression (Fig. S3E), this may be simply caused by potential difference of FGF4 protein amount between D0 and D1. Adding some words in the manuscript would be helpful to understand the relationship between initiation of differentiation by withdrawing LIF or FBS and phosphorylation of ERK1/2 and Yy1. Is it possible to add some speculation in Discussion about this missing link, how differentiation induces Yy1 activation?

4. Lines 137-139

As FGF2 is not presented in Appendix Fig. S1B,C, FGF2 should be shown in this figure.

5. Figure 2A-E

It is not clearly described at which time point and how long the inhibitors were added in these experiments. If the inhibitors were added from the beginning of differentiation, it has to be clearly mentioned in the figure legend or Methods section.

6. Figure 3A

I thank the authors for adding this data. Is it possible to make a quantification of the pYY1 band intensity? In the text, it is not clearly described when pYY1 accumulated in this analysis. It appears increasing at D1, but not clear. I recommend the authors to describe this point clearly with quantitated values in the text.

7. Figure 3B-G

Which state of ESCs was used in these WB and ChIP experiments, undifferentiated cells or early differentiating cells (if so, which day of differentiation)? I could not find any description. There are strong pYY1 and pERK bands in control ESCs, which look like D1 differentiating cells shown in Fig 3A and Fig S4BC. Many data in the manuscript still lack detailed explanation of the experimental condition in figure legend or Method section making difficult to understand and evaluate the presented data.

8. Figure 4G,H

These data also lack the information of experiment about how long the cells were differentiated.

9. Line 346

Which ESC line was used in this study? I asked the same question in the original manuscript but can't seem to find the answer in the revised manuscript. In lines 433 and 436, PGK12.1 suddenly appeared. Is it PGK12.1 provided from Neil Brockdorff and used in all experiments? If so, "PGK12.1" should be first described at around the line 346.

Points to be corrected

1. Appendix Fig. S3E

As Xist is not mRNA, the Y axis should be "Relative expression of Xist".

2. Figure 3I

pCAGGS-Yy1-S120D is not correctly presented with "+".

3. Lines 57-58 "the loss of pluripotency derepresses Xist upregulation"

The word "upregulation" should be removed from this sentence.

4. Line 197

"the phosphorylated levels of both YY1 and phosphorylated ERK" should be "the phosphorylated levels of both YY1 and ERK".

As I can still find writing mistakes, the manuscript needs to be carefully checked by a native English speaker.

5. Line 217

"Appendix Fig. S4D" should be "Appendix Fig. S5D".

Referee #1:

The authors have done a good job in addressing the reviewers' requests, mine included. Yet, their main claim on how the FGF4 pathway regulates Xist expression (and XCI) via both YY1 phosphorylation and pluripotency factors still lacks substantial back-up, especially regarding the YY1 phosphorylation.

Response: We thank the reviewer for the positive evaluation and constructive suggestions. The manuscript has been revised for clarity, with added methodological details and improved figure consistency. Detailed responses are presented below.

1/ The new data presented with the YY1 mutants is still not sufficiently convincing, since the effects on Xist expression levels are relatively low. For instance, it is not clear

whether *Xist* levels upon overexpression of the no-phosphorylation mutants are significantly different from the phosphorylation-mimic mutants (it does not seem to be the case). It might be that the effects would only be observed in a background without wildtype YY1, as suggested by the other reviewer. The authors report that YY1-knockout ES cells are not viable, so probably a YY1-degron would be necessary to address this point fully.

Response: We appreciate the reviewer's concern that the effects of YY1 phosphorylation on *Xist* expression. In response to your concerns, we have given the following explanation and added some new data:

Regarding the concern about "the effects on *Xist* expression levels are relatively low", it should be noted that *Xist* is co-regulated by a series of activating and repressing factors (Jonkers et al, 2009; Makhlof et al, 2014; Navarro et al, 2008; Ravid Lustig et al, 2023; Tian et al, 2010). Thus, it seems explainable that the change in *Xist* following overexpression of wild-type and mutant *Yy1*, despite its critical role in inducing XCI initiation, is limited. As the results of a previous study has shown *Yy1* knockdown results in a decrease of approximately 1/2 to 3/4 of the *Xist* (Makhlof et al, 2014).

More importantly, **we agree with your point that the effects of exogenous *Yy1* overexpression could be masked by the presence of endogenous *Yy1***. To exclude this possibility, we generated an acute YY1 depletion system by introducing an inducible degron tag (Li et al, 2023; Weintraub et al, 2017) into the endogenous *Yy1* locus in ES cells. This allowed us to remove endogenous YY1 protein within 24 hours of degron activation, thereby creating a background free of wild-type YY1 (**see below, Figure 1A-C**) while maintaining short-term cell viability (complete *Yy1* knockout severely impairs cell viability and proliferation, as mentioned in our previous round of revisions). Based on the YY1-depleted background, we then transfected phosphorylation deficient (S→A) and phospho-mimic (S→D) *Yy1* mutants into YY1-degron ES cells after acute depletion of endogenous YY1. The phospho-mimic mutantation (S→D) resulted in a substantial and sustained *Xist* upregulation compared to the non-phosphorylation mutantation (S→A) (**see below, Figure 1D**). Thus, under the premise of excluding potential interference from endogenous YY1, our new data indicate that phosphorylation status of YY1 can directly modulate *Xist* transcriptional activation.

We have added these new data into the Results section (**Lines 213–220**) and revised the Discussion (**Lines 296-305**) to reflect that the acute removal of endogenous YY1 supports the model in which YY1 phosphorylation is a contributing, but not exclusive, mechanism for *Xist* upregulation during random XCI initiation.

Figure 1. (A) Western blot analysis of YY1 protein levels in wild-type ES cells and YY1-dTAG ES cells at 24h after dTAG treatment or 24 hours after dTAG withdrawal. β -TUBULIN is used as a loading control. (B) Schematic showing the differentiation protocol for YY1-dTAG ES cells with DMSO or dTAG treatment. (C) Relative *Xist* expression levels between DMSO- and dTAG-treated ES cells at different time points, revealing the functional loss of wild-type endogenous YY1; $n=3$. (D) Relative expression levels of *Xist* in YY1-dTAG ES cells transfected with an empty vector (pCAGGS), or overexpression vector of WT or mutant *Yy1*; $n=4$. (C and D) Data are shown as means \pm SEM, P value was calculated by two-way ANOVA test with multiple comparisons (C) and one-way ANOVA test with multiple comparisons (D).

2/ The data presented to justify their claim is mostly correlative and lacks further controls. Examples: the increase in YY1-phosphorylation coinciding with the time window of XCI-induction (D1 of differentiation); is this effect specific to XX cells? The lower binding of YY1 to the *Xist* promoter in *Fgf4*-KO - is this specific to the *Xist* promoter, or is there lower binding in general for YY1 sites that are more differentiation-specific? (and would also be affected given the altered differentiation status of *Fgf4*-KO)

Response: Thank you for your constructive comments, which helps us improve the data. We have added a series of new data to address your concerns.

(1) Regarding your first concern. To determine whether YY1 phosphorylation is restricted to the initiation stage in female ES cells, we analyzed the phosphorylation status of YY1 during the differentiation of male ES cells (R1) under same differentiation condition. We observed that although R1 displayed steady-state levels of total YY1 upon XCI initiation, similar to that in female ES cells (PGK12.1), the change in YY1 phosphorylation, detected via both Phos-tag assays and Western blot using a specific antibody against phosphorylated YY1, was not obvious. By contrast, when detected with this specific antibody, PGK12.1 cells showed an increase in YY1 phosphorylation during differentiation (see below Figure 2), consistent with the result obtained from

Phos-tag assays in our previous round of revisions (Figure 3A in the previous manuscript). Although there are some differences in the results between different methods, they consistently suggest that **the increased YY1 phosphorylation during early stage of differentiation is prone to occur in female ES cells** (see below Figure 2). We have added these new data to the Results and Discussion sections in the revised manuscript (Appendix Fig. S5D, E; Lines 186-189, Lines 284-295).

Figure 2. (A) Phos-tag analysis of pYY1 and Western blot analysis of β -TUBULIN in male ES cells (R1) on day 0-0h, day 0-2h, day 1, day 3 and day 5 of differentiation. (B) Western blot analysis of p-YY1(Ser365) and total YY1 in male ES cells (R1) (upper panel) on day 0-0h, day 0-2h, day 1, day 3 and day 5 of differentiation and female ES cells (PGK12.1) (lower panel) on day 0, day 1, day 3 and day 5 of differentiation.

(2) Regarding your second concern, to address whether the reduced YY1 binding in *Fgf4* KO-84# ES cells is specific to the *Xist* promoter or reflects a broader loss at differentiation-associated YY1 sites, we performed CUT&Tag at Day1 of differentiation, at which the YY1 phosphorylation is most notable, and compared global occupancy of YY1 between wild-type and *Fgf4* KO-84# ES cells. Our results showed that a considerable number of peaks in wild-types cells lost their YY1 binding due to *Fgf4* deficiency (see below Figure 3), suggesting that the YY1 binding at these sites largely depends on FGF4 signaling and FGF4-induced YY1 binding are not specific to *Xist* promoter. We have added these new data in the revised manuscript (Figure 3, Lines 195-201, Lines 296-305).

Figure 3. (A) Scatter plots comparing the YY1 CUT&Tag signals (entire genome) between replicates (two biological replicates) for each stage. The Pearson correlation coefficients are also shown. (B) Average profile (metaplot) of normalized YY1 CUT&Tag signal centered on transcription start sites in WT and *Fgf4* KO-84# ES cells. Signals were plotted across a ± 3 kb window around the TSS. The x-axis indicates distance from the TSS (kb), and the y-axis indicates normalized YY1 enrichment. (C) Heatmaps of YY1 binding enrichment and corresponding average CUT&Tag signal profiles in WT and *Fgf4* KO-84# ES cells. CUT&Tag signals are plotted around TSS (± 3 kb) associated with YY1 peaks. YY1 peaks are categorized as gained ($n = 187$), lost ($n = 191$), or shared ($n = 1,809$) in *Fgf4* KO-84# compared with WT. (D,E) The UCSC browser view showing YY1 enrichment at representative promoters region in WT, *Fgf4* KO-84# ES cells. Gene models are shown below; arrows indicate the direction of transcription. (F) CUT&RUN-qPCR enrichment of YY1 at the *Yy1*, *Xist*, *Klf5*, *Pou5f1* locus is shown as fold enrichment over IgG in WT and *Fgf4* KO-84#

ES cells; $n=3$. Data are shown as means \pm SEM, P value was calculated by unpaired two-tailed Student's t -test.

3/ The experiments with retinoic acid (RA) seem to suggest that the effects of *Fgf4*-KO on *Xist* expression are not solely due to impaired differentiation (thus highlighting other mechanisms involved, indirectly supporting the YY1-phosphorylation hypothesis). However, the authors do not show whether in the RA treatment for *Fgf4*-KO 84# ES cells the levels of pluripotency factors are sufficiently low, similar to wildtype or to when FGF4 is supplemented (and rescues) *Xist* expression. In other words, could it be that the RA treatment does not rescue *Xist* expression levels because the pluripotency factors are still too high?

Response: We thank the reviewer for this insightful comment to refine our experimental design. As you have pointed out, it is indeed an important point whether RA treatment sufficiently reduced pluripotency factors in *Fgf4*-KO 84# ES cells to levels comparable to those in wild-type ES cells. Based on your comments, we have detected the expression levels of pluripotency markers that showed a delayed decline in *Fgf4* KO-84# ES cells, *i.e.*, *Nanog*, *Prdm14* and *Rex1* under RA treatment (see below Figure 4). Our data showed that these factors were significantly downregulated by RA induction in *Fgf4* KO-84# ES cells, to levels comparable to or close to those in wild-type ES cells. Thus, the lack of *Xist* rescue in the RA-treated *Fgf4*-KO ES cells could not be solely attributed to impaired differentiation or delayed decline of some core pluripotency factors. This supports the notion that additional mechanisms, such as impaired signaling pathways (potentially involving YY1 phosphorylation and other unidentified mechanisms), contribute to the failure of *Xist* upregulation in *Fgf4* KO-84# ES cells. We have included these new data and added relevant statements in the revised manuscript (Appendix Fig. S7B and Lines 228-233).

Figure 4. Dynamic expression levels of *Nanog* (left), *Prdm14* (medium), *Rex1* (right) in WT, *Fgf4* KO-84#, *Fgf4* KO-84#+RA ES cells at days 0, 1, 3 and 5 of differentiation; $n = 3$. The schematic diagram below the

X-axis indicates the timeline of differentiation. The day when ES cells were transferred to the differentiation medium was designated as day 0. Data are shown as means \pm SEM, compared with the wild type ES cells.

4/In the absence of further evidence, the authors could still present their data as it is but would have to drop the claim about YY1 phosphorylation and change the manuscript accordingly.

Response: Thanks for your insightful comments that make our statement more rigorous. We have carefully refined the relevant statements to avoid overstatements. In addition, we have added new data based on your comments, including: (1) overexpression experiments showing the effect of YY1 in different phosphorylation states on *Xist* expression on a endogenous YY1-depleted background; (2) the phosphorylation status of YY1 of female and male ES cells during differentiation; (3) the genome-wide change of YY1 binding due to *Fgf4* deficiency. We have added relevant descriptions to the Discussion section to provide a deeper understanding of the role of YY1 phosphorylation in *Xist* upregulation (Lines284-295).

The manuscript still needs substantial editing for clarity and consistency; maybe this can be provided by the journal.

Response: Thank you for your reminder. We have carefully edited the manuscript and optimized the expression to enhance clarity and accuracy, in accordance with the journal's requirements.

Referee #2:

I thank the authors to have responded to most of my concerns. I think the revised manuscript has much more clear messages for readers. I appreciate that they made efforts to strengthen the data supporting phosphorylated Yy1 could up-regulate *Xist* (Figure 3). I still have several minor comments listed below about the newly added data in the revised manuscript. As the manuscript still contains lack of experimental conditions in several figures, e.g. Figures 3 and 4, the authors should carefully check the figure legend and Methods section.

1. Lines 131-135

"More importantly, we found that the addition of exogenous FGF4 to the differentiation medium, which compensates for the endogenous loss of FGF4, completely rescued *Xist* upregulation and coating, as well as the H3K27me3 state, to levels comparable to those in wild-type ES cells (Fig 1I,J; Appendix Fig. S3A)." Does Appendix Fig. S3A

correspond to this result? It looks just showing the lack of ERK1/2 phosphorylation in FGF4 KO ESC lines. If so, this data should be placed in Fig S2, between the panels C and D.

Response: We apologize for our ambiguous statements that led to your misunderstanding. Appendix Fig. S3A in the original manuscript does not show the rescue of *Xist* upregulation and H3K27me3 by exogenous FGF4. Instead, the presentation of the absence of ERK1/2 phosphorylation in the *Fgf4* KO-84# ES cells during differentiation, is to functionally confirm that FGF4 has been knocked out in these cells, because FGF-triggered ERK activation is the hallmark of early differentiation of ES cells (Kunath et al, 2007; Lanner & Rossant, 2010).

In our figure layout, Appendix Fig. S2C,D demonstrate that *Fgf4* is successfully knocked out in the *Fgf4* KO-84# ES cells at the mRNA and protein levels in the ground state, whereas the Appendix Fig. S3A shows that these cells fail to activate ERK during differentiation, consistent with the dependence of ERK activation on FGF signaling. Thus, these datasets together confirm both the molecular and functional loss of FGF4. To avoid confusion, we have now

1. **Corrected the text** so that the rescue of *Xist* and H3K27me3 upon addition of exogenous FGF4 is only supported by Fig. 1I,J, and the Figure citation to Appendix Fig. S3A has been removed from this sentence (**Line 124**).
2. **Relocated the ERK1/2 phosphorylation data** from Appendix Fig. S3A to the position immediately following Fig. S2 C and D, as a new panel between the original panels D and E (**now Appendix Fig. S2C–E**), in line with the reviewer's suggestion. In addition, we have refined the statements in the figure legend to make our intentions clearer and easier to understand: "showing a functional knockout of *Fgf4*" has been change to "functionally confirming that FGF4 has been knocked out because FGF-triggered ERK activation is the hallmark of early differentiation of ES cells (Kunath et al, 2007; Lanner & Rossant, 2010)". To make our intentions clearer and easier to understand.

We hope this reorganization and clarification resolve the confusion.

2. Fig. 3H-K

I thank the authors for adding these new data using phospho-mimic Yy1 and endogenous *Xist* promoter. Although the results support the significant role of Yy1 phosphorylation to up-regulate *Xist*, the degree of up-regulation (~2-fold) is weaker than endogenous *Xist* up-regulation observed in differentiating ESCs (usually > 10-fold upon differentiation). This would suggest the necessity of other factor(s) other than phosphorylated Yy1 to fully up-regulate *Xist*. I recommend the authors to add a few sentences to describe this point in the text.

Response: Thanks so much for your detailed and insightful comments to improve our discussion section. As you have pointed out, *Xist* is co-regulated by a series of activating and repressing factors (Jonkers et al, 2009; Makhlof et al, 2014; Navarro et al, 2008; Ravid Lustig et al, 2023; Tian et al, 2010). Thus, it seems explainable that the change in *Xist* following overexpression of wild-type and mutant *Yy1*, despite its critical role in inducing XCI initiation, is limited. As the results of a previous study has shown, *Yy1* knockdown resulted in a decrease of approximately 1/2 to 3/4 of the *Xist*(Makhlof et al, 2014). We have added relevant descriptions to the Discussion section to explain the relatively small changes in our results (Lines 213-220).

3. Appendix Fig. S1B, S3A,E and S4B,C

I thank the authors to add *Yy1* phosphorylation time-course data (Fig 3A). This raises a new question on how *Yy1* is phosphorylated only upon differentiation. FGF4 is already expressed in undifferentiated WT ESCs (Fig S1B 0 hour, Fig S1D D0) but does not induce *Xist* up-regulation, which is consistent with the fact that ERK1/2 are not yet phosphorylated in undifferentiated state (Fig S4B,C). ERK1/2 are, however, rapidly phosphorylated in 0.5 hour upon differentiation (Fig S4B,C), which is followed by phosphorylation of *Yy1* on day 1 of differentiation (Fig 3A). These observations indicate the importance of differentiation itself to induce a rapid ERK1/2 phosphorylation. How this rapid phosphorylation of ERK1/2 upon differentiation could happen? As the exogenous addition of FGF4 increases *Xist* expression (Fig. S3E), this may be simply caused by potential difference of FGF4 protein amount between D0 and D1. Adding some words in the manuscript would be helpful to understand the relationship between initiation of differentiation by withdrawing LIF or FBS and phosphorylation of ERK1/2 and *Yy1*. Is it possible to add some speculation in Discussion about this missing link, how differentiation induces *Yy1* activation?

Response: Thanks so much for your insightful comments to enhance our understanding and fill the gap between differentiation initiation and phosphorylation of ERK-YY1 axis. As you have pointed out, despite the high-level expression of *Fgf4* in undifferentiated ES cells, ERK is not fully phosphorylation due to several "safety mechanisms", such as (1) FGF-induced ERK phosphorylation can be inhibited by LIF via binding of ERK and Sprouty2(Liu et al, 2013); (2) upregulated dual-specificity phosphatase 9 (DUSP9) attenuates ERK activation(Li et al, 2012) et al. Based on these information, as well as the temporal gap between ERK activation and YY1 phosphorylation, it can be speculated that upon LIF withdrawal, FGF-ERK signaling is activated, which in turn initiates the transition from a pluripotent state toward differentiation. Subsequently, the pre-existing YY1 is phosphorylated under the induction of ERK, thereby activating its transcriptional activity. We have added these

ideas to the discussion section of the revised manuscript (Lines 296-305).

4. Lines 137-139

As FGF2 is not presented in Appendix Fig. S1B,C, FGF2 should be shown in this figure.

Response: Thanks for your detailed reminder to make our results much clearer and easier to understand. In our previous round of revisions, we did not include the *Fgf2* expression in Appendix Fig. S1B, C, because *Fgf2* expression is undetectable during embryonic stem cell differentiation, as shown by publicly available datasets (Barros De Andrade E Sousa et al, 2019; Schulz et al, 2014). Following your comments, we have now presented the expression levels (assigned as 0) of *Fgf2* to Appendix Fig. S1B, C in the revised version. The figure and legend have been updated accordingly (see below Figure 5, Appendix Fig. S2B,C).

Figure 5. Dynamic expression levels of *Fgfs* (upper panel), *Fgfrs* (upper panel) and interaction score (down panel) at different time points of differentiation using publicly available transcriptome data (Barros De Andrade E Sousa et al, 2019; Schulz et al, 2014).

5. Figure 2A-E

It is not clearly described at which time point and how long the inhibitors were added in these experiments. If the inhibitors were added from the beginning of differentiation, it has to be clearly mentioned in the figure legend or Methods section.

Response: We thank the reviewer for pointing this out. In all experiments shown in this figure, the inhibitors were added at the onset of differentiation (day 0) and maintained in the culture medium until sample collection, unless otherwise indicated. We have now clarified this information in both the figure legend and the Methods

section, which state that ES cells were differentiated in the presence of the indicated inhibitors from day 0 to day 5. In addition, to make it easier to understand, we have added a schematic diagram to illustrate the experimental workflow (see below Figure 6, Figure. 2A, Lines 760-763).

Figure 6. Schematic of inhibitor treatment during differentiation from D0 to D5 (upper panel), and Immunofluorescence staining of H3K27me3 and *Xist* FISH staining (down panel) in WT ES cells treated with the inhibitor specific to PLC γ , STAT, PI3K and MEK/ERK on day 5 of differentiation; PLC γ i: U73122 (U), 5 μ M; STATi: Fludarabine (Flu), 1 μ M plus SH-4-54 (SH), 1 μ M; PI3Ki: LY294002 (LY) 1 μ M; MEK/ERKi: PD0325901 (PD), 1 μ M. In the upper panels, the white arrow in the upper panels denotes a nuclear *Xist* cloud. % *Xist*-cloud = number of cells with *Xist* cloud / total number of analyzed cells. Cells with one *Xist* cloud signal were considered as normal XCI and included in the statistical unit. In the lower panels, the white in the nuclear H3K27me3 domain. % H3K27me3 = number of cells with nuclear H3K27me3 domains/ total number of analyzed cells. Cells with one H3K27me3 domain were considered as normal XCI and included in the statistical unit.

6. Figure 3A

I thank the authors for adding this data. Is it possible to make a quantification of the pYY1 band intensity? In the text, it is not clearly described when pYY1 accumulated in this analysis. It appears increasing at D1, but not clear. I recommend the authors to describe this point clearly with quantitated values in the text.

Response: Thanks for this insightful comment. We have adopted your suggestion and performed a quantitative analysis of the pYY1 band intensity and added the results into the revised manuscript (see below Figure 7, Figure 3A). The data show that the phosphorylation level of YY1 gradually increased 2h after induction of differentiation

and peaked on day 1. We have revised the Results section (Lines 183–185) and the figure legend to clearly describe this dynamic change in YY1 phosphorylation.

Figure 7. Phos-tag analysis of pYY1 and Western blot analysis of β -TUBULIN in WT ES cells at day0-0h, day0-2h, day 1, day 3 and day 5 of differentiation (left), along with the quantitative analysis of pYY1 phosphorylation levels (right), $n=4$. Data are shown as means \pm SEM, P value was calculated by two-way ANOVA test with multiple comparisons.

7. Figure 3B-G

Which state of ESCs was used in these WB and ChIP experiments, undifferentiated cells or early differentiating cells (if so, which day of differentiation)? I could not find any description. There are strong pYY1 and pERK bands in control ESCs, which look like D1 differentiating cells shown in Fig 3A and Fig S4BC. Many data in the manuscript still lack detailed explanations of the experimental condition in figure legend or Method section making difficult to understand and evaluate the presented data.

Response: We thank the reviewer for this suggestion. The duration of cell differentiation and the sample collection time have been added to the legend, as follows: (B) Phos-tag analysis of pYY1 and Western blot analysis of pERK, ERK in WT, *Fgf4* KO-84#, *Fgf4* KO-84#+FGF4 ES cells on day 0-2h of differentiation; FGF4 10 ng/mL. (C and D) Co-IP assays (C) and quantification (D) of interaction between YY1 and pERK, in differentiating WT, *Fgf4* KO-84#, and *Fgf4* KO-84#+FGF4 ES cells on day 1 of differentiation; $n = 3$. (E) Western blot analysis of pYY1, pERK, ERK in WT treated with or without MEK/ERKi on day 0-2h of differentiation. MEK/ERKi, PD0325901, 1 μ M. (F) ChIP analysis of YY1 enrichment at the binding site within the *Xist*'s 5' region in WT, *Fgf4* KO-84#, and WT ES cells treated with MEK/ERKi on day 1 of differentiation; $n = 3$. (G) ChIP analysis of YY1 enrichment at the binding site within the *Xist*'s 5' region in *Fgf4* KO-84# after FGF4 addition on day 1 of differentiation; $n = 3$ (Lines 807-815).

8. Figure 4G,H

These data also lack the information of experiment about how long the cells were differentiated.

Response: We thank the reviewer for this suggestion. The duration of cell differentiation and the sample collection time have been added to the figure legend, as follows: Immunofluorescence staining (G) and quantification (H) of H3K27me3 domains in WT, *Fgf4* KO, *Fgf4-Nanog* DKO, *Fgf4-Prdm14* DKO, *Fgf4-Rex1* DKO ES cells on day 5 of differentiation (Lines 853-855).

9. Line 346

Which ESC line was used in this study? I asked the same question in the original manuscript but can't seem to find the answer in the revised manuscript. In lines 433 and 436, PGK12.1 suddenly appeared. Is it PGK12.1 provided from Neil Brockdorff and used in all experiments? If so, "PGK12.1" should be first described at around the line 346.

Response: We thank the reviewer for pointing out this omission. We confirm that *PGK12.1* ES cells (a gift from Prof. Neil Brockdorff) were used in all experiments in this study. To clarify, we have now introduced and described *PGK12.1* ES cells at their first appearance (around Lines 362-363) in the revised manuscript.

Points to be corrected

1. Appendix Fig. S3E

As *Xist* is not mRNA, the Y axis should be "Relative expression of *Xist*".

Response: Thanks so much for your reminder. The Y-axis label in Appendix Fig. S3D (formerly S3E) has been corrected to "Relative expression of *Xist*".

2. Figure 3I

pCAGGS-Yy1-S120D is not correctly presented with "+".

Response: Thanks so much for your reminder. The notation in Fig. 3L (formerly 3I) has been corrected to replace "+" with "-" for pCAGGS-Yy1-S120D.

3. Lines 57-58 "the loss of pluripotency derepresses *Xist* upregulation"

The word "upregulation" should be removed from this sentence.

Response: The word "upregulation" has been removed from the sentence in Lines 48–49.

4. Line 197

"the phosphorylated levels of both YY1 and phosphorylated ERK" should be "the phosphorylated levels of both YY1 and ERK". As I can still find writing mistakes, the manuscript needs to be carefully checked by a native English speaker.

Response: The sentence in Line 197 has been corrected to "the phosphorylated levels of both YY1 and ERK". In addition, the manuscript has been carefully re-checked to eliminate remaining language errors.

5. Line 217

"Appendix Fig. S4D" should be "Appendix Fig. S5D".

Response: The figure citation in Line 217 has been corrected to "Appendix Fig. S5D".

References:

- Barros De Andrade E Sousa L, Jonkers I, Syx L, Dunkel I, Chaumeil J, Picard C, Foret B, Chen C, Lis JT, Heard E, Schulz EG, Marsico A (2019) Kinetics of Xist-induced gene silencing can be predicted from combinations of epigenetic and genomic features. *GENOME RES* **29**(7): 1087-1099
- Jonkers I, Barakat TS, Achame EM, Monkhorst K, Kenter A, Rentmeester E, Grosveld F, Grootegoed JA, Gribnau J (2009) RNF12 is an X-Encoded dose-dependent activator of X chromosome inactivation. *CELL* **139**(5): 999-1011
- Kunath T, Saba-El-Leil MK, Almousailleakh M, Wray J, Meloche S, Smith A (2007) FGF stimulation of the Erk1/2 signalling cascade triggers transition of pluripotent embryonic stem cells from self-renewal to lineage commitment. *DEVELOPMENT* **134**(16): 2895-2902
- Lanner F, Rossant J (2010) The role of FGF/Erk signaling in pluripotent cells. *Development (Cambridge, England)* **137**(20): 3351-3360
- Li L, Lai F, Hu X, Liu B, Lu X, Lin Z, Liu L, Xiang Y, Frum T, Halbisen MA, Chen F, Fan Q, Ralston A, Xie W (2023) Multifaceted SOX2-chromatin interaction underpins pluripotency progression in early embryos. *Science (New York, N.Y.)* **382**(6676): eadi5516
- Li Z, Fei T, Zhang J, Zhu G, Wang L, Lu D, Chi X, Teng Y, Hou N, Yang X, Zhang H, Han JJ, Chen Y (2012) BMP4 Signaling Acts via dual-specificity phosphatase 9 to control ERK activity in mouse embryonic stem cells. *CELL STEM CELL* **10**(2): 171-182
- Liu J, Hsu Y, Kao C, Su H, Chiu I (2013) Leukemia inhibitory factor-induced Stat3 signaling suppresses fibroblast growth factor 1-induced Erk1/2 activation to inhibit the downstream differentiation in mouse embryonic stem cells. *STEM CELLS DEV* **22**(8): 1190-1197
- Makhlouf M, Ouimette J, Oldfield A, Navarro P, Neuillet D, Rougeulle C (2014) A prominent and conserved role for YY1 in Xist transcriptional activation. *NAT COMMUN* **5**: 4878
- Navarro P, Chambers I, Karwacki-Neisius V, Chureau C, Morey C, Rougeulle C, Avner P (2008) Molecular coupling of Xist regulation and pluripotency. *Science (New York, N.Y.)* **321**(5896): 1693-1695
- Ravid Lustig L, Sampath Kumar A, Schwämmle T, Dunkel I, Noviello G, Limberg E, Weigert R, Pacini G, Buschow R, Ghauri A, Stötzl M, Wittler L, Meissner A, Schulz EG (2023) GATA transcription factors drive initial Xist upregulation after fertilization through direct activation of long-range enhancers. *NAT CELL BIOL* **25**(11): 1704-1715

Schulz EG, Meisig J, Nakamura T, Okamoto I, Sieber A, Picard C, Borensztein M, Saitou M, Blüthgen N, Heard E (2014) The two active X chromosomes in female ESCs block exit from the pluripotent state by modulating the ESC signaling network. *CELL STEM CELL* **14**(2): 203-216

Tian D, Sun S, Lee JT (2010) The long noncoding RNA, Jpx, is a molecular switch for X chromosome inactivation. *CELL* **143**(3): 390-403

Weintraub AS, Li CH, Zamudio AV, Sigova AA, Hannett NM, Day DS, Abraham BJ, Cohen MA, Nabet B, Buckley DL, Guo YE, Hnisz D, Jaenisch R, Bradner JE, Gray NS, Young RA (2017) YY1 Is a Structural Regulator of Enhancer-Promoter Loops. *CELL* **171**(7): 1573-1588

Dear Dr Tian,

Thank you for submitting the revised version of your manuscript. I have now evaluated your amended manuscript and concluded that the remaining minor concerns have been sufficiently addressed.

I am thus pleased to inform you that your manuscript has been accepted for publication in the EMBO Journal.

Kind regards,

Daniel Klimmeck

Daniel Klimmeck, PhD
Senior Editor
The EMBO Journal
EMBO
Postfach 1022-40
Meyerohofstrasse 1
D-69117 Heidelberg
contact@embojournal.org

Please note that it is The EMBO Journal policy for the transcript of the editorial process (containing referee reports and your response letters) to be published as an online supplement to each paper. If you should prefer removal of any referee-only figures included in the point-by-point response(s), e.g. because they may still be used for future publication or because they have been reproduced from published work by others, please do let us know immediately via response email. More information is available here: <https://link.springer.com/partners/embo-press/editorial-policies#Peer%20review>